# Monosynaptic premotor circuit tracing reveals neural substrates for oro-motor coordination

**Edward Stanek IV[1], Steven Cheng[1], Jun Takatoh[1], Bao-Xia Han[1], Fan Wang[1,2]***

[1]Department of Neurobiology, Duke University Medical Center, Durham, United States; [2]Department of Cell Biology, Duke University Medical Center, Durham, United States

**Abstract** Feeding behaviors require intricately coordinated activation among the muscles of the jaw, tongue, and face, but the neural anatomical substrates underlying such coordination remain unclear. In this study, we investigate whether the premotor circuitry of jaw and tongue motoneurons contain elements for coordination. Using a modified monosynaptic rabies virus-based transsynaptic tracing strategy, we systematically mapped premotor neurons for the jaw-closing masseter muscle and the tongue-protruding genioglossus muscle. The maps revealed that the two groups of premotor neurons are distributed in regions implicated in rhythmogenesis, descending motor control, and sensory feedback. Importantly, we discovered several premotor connection configurations that are ideally suited for coordinating bilaterally symmetric jaw movements, and for enabling co-activation of specific jaw, tongue, and facial muscles. Our findings suggest that shared premotor neurons that form specific multi-target connections with selected motoneurons are a simple and general solution to the problem of orofacial coordination.

***For correspondence:** fan.wang@duke.edu

**Competing interests:** The authors declare that no competing interests exist.

**Reviewing editor**: Peggy Mason, University of Chicago, United States

## Introduction

Behaviors are executed through coordinated activity of multiple groups of motor neurons and their muscle targets. Coordination of jaw and tongue muscles during feeding behaviors represents one of the most intricate mechanisms of the motor system and has been observed in a wide range of animals including humans (*Gerstner and Goldberg, 1991*; *Thexton and McGarrick, 1994*; *Takada et al., 1996*; *Palmer et al., 1997*; *Ishiwata et al., 2000*; *Miller, 2002*). Here, coordination concerns primarily the adjustment of both timing and sequence of muscle activation to enable smooth, effective jaw and tongue movements. Three basic forms of coordination are consistently observed in feeding behaviors. First, the left and right jaw muscle activities are temporally symmetric, which is necessary because the mandible is joined by ligaments at the midline. Second, during chewing the activity of the tongue and jaw muscles is held to a similar low frequency rhythm, with the tongue positioning food between the surfaces of the teeth while the jaw moves the teeth to break down food (*Thexton and McGarrick, 1994*; *Takada et al., 1996*; *Hiyama et al., 2000*; *Naganuma et al., 2001*; *Yamamura et al., 2002*). Third, during chewing (*Gerstner and Goldberg, 1991*; *Liu et al., 1993*; *Naganuma et al., 2001*), licking (*Travers et al., 1997*), and suckling (*Thexton et al., 1998*), the tongue-protruding and jaw-opening muscles are co-active during jaw opening, while tongue-retracting and jaw-closing muscles are co-active during jaw closing. This co-activation also occurs in cortically-induced fictive mastication and occurs regardless of the frequency of cortical stimulation or the intensity of sensory stimuli applied (*Gerstner and Goldberg, 1991*; *Liu et al., 1993*).

The neural architecture enabling these different yet specific forms of jaw-tongue-facial muscle coordination remain unclear. In the more extensively studied vertebrate spinal cord, the network that

**eLife digest** Chewing requires highly coordinated movements of the tongue and jaw. The tongue pushes food around the mouth, keeping it within reach of the teeth, while rhythmic movements of the jaw enable the teeth to grind up food without injuring the tongue itself. However, despite the importance of tongue–jaw coordination, the neuroanatomical circuits that underlie it have not been studied in detail. Stanek et al. have now mapped these neural circuits using a cutting edge tracing technique in mice.

Muscles are activated by signals from neurons called motoneurons, which are themselves activated by signals from so-called premotor neurons. To identify the neural circuits responsible for tongue–jaw coordination, Stanek et al. injected a modified version of the rabies virus into the muscles that move the jaw and/or the tongue. This modified virus, which was also labeled with a fluorescent protein, was able to jump 'backwards' across the junctions between the muscles and the motoneurons, and then back to the premotor neurons. The fluorescent label allowed the neural circuits to be visualized under a fluorescence microscope.

It had been assumed that distinct populations of premotor neurons would control the activity of different muscles. However, when viruses labeled with red fluorescent protein were injected into the muscles on the left side of the jaw, while viruses with green labels were injected into the muscles on the right side, a number of premotor neurons were found to display both red and green fluorescence. This indicates that some premotor neurons control muscles on both sides of the jaw, providing an effective means of coordinating bilateral muscle activity. A similar sharing of premotor neurons was observed between motoneurons that regulate jaw opening and those that trigger tongue protrusion, and between those that regulate jaw closing and tongue retraction.

As well as providing new insights into the neuronal circuits that control the movements of the jaw and tongue, the work of Stanek et al. may have identified a general principle—namely the sharing of premotor neurons—that could be common to other circuits that produce coordinated muscle activity.

generates the coordinated and rhythmic muscle activity during locomotion is referred to as the central pattern generator (CPG). Separate CPGs are known to control separate limb muscles, with forelimb CPGs located in the cervical enlargement and hindlimb CPGs residing in the lumbar enlargements (*Kiehn, 2011*). Limb coordination occurs through interactions between these separate CPGs (*Kiehn, 2011*). By analogy, it is conceivable that different orofacial muscles are also controlled by different CPGs with their interaction resulting in orofacial coordination, although the evidence for well-defined jaw, face, and tongue CPGs is lacking. On the other hand, previous studies injecting two different retrograde tracers into two different cranial motor nuclei have suggested the existence of neurons projecting to both nuclei (*Amri et al., 1990*; *Li et al., 1993*; *Kamogawa et al., 1994*; *Popratiloff et al., 2001*; *Kondo et al., 2006*). However, due to limitations of the neural tracer technique, such as non-specific labeling of passing fibers and labeling of the entire nucleus rather than the motoneurons innervating specific muscles, whether there are common premotor neurons simultaneously innervating specific motoneuron groups enabling the above mentioned orofacial coordination remained unclear. In this study, we employed a recently established monosynaptic circuit tracing methodology to identify premotor neurons of the jaw-closing masseter and tongue-protruding genioglossus motoneurons (See 'Materials and methods'). Analysis of the resultant labeling reveals several premotor circuit elements that are well suited for the orofacial coordination observed in feeding behaviors.

## Results

### Transsynaptic mapping of premotor circuitry controlling the jaw-closing masseter motoneurons

As described in the 'Materials and methods' and schematically illustrated in *Figure 1*, we have developed a mouse line, Chat::Cre; RΦGT, that enables us to inject deficient rabies virus (ΔG-RV) into desired muscles resulting in transsynaptic tracing of the corresponding premotor circuitry in neonatal mice (*Takatoh et al., 2013*). For further description and discussion of this method, see the 'Materials

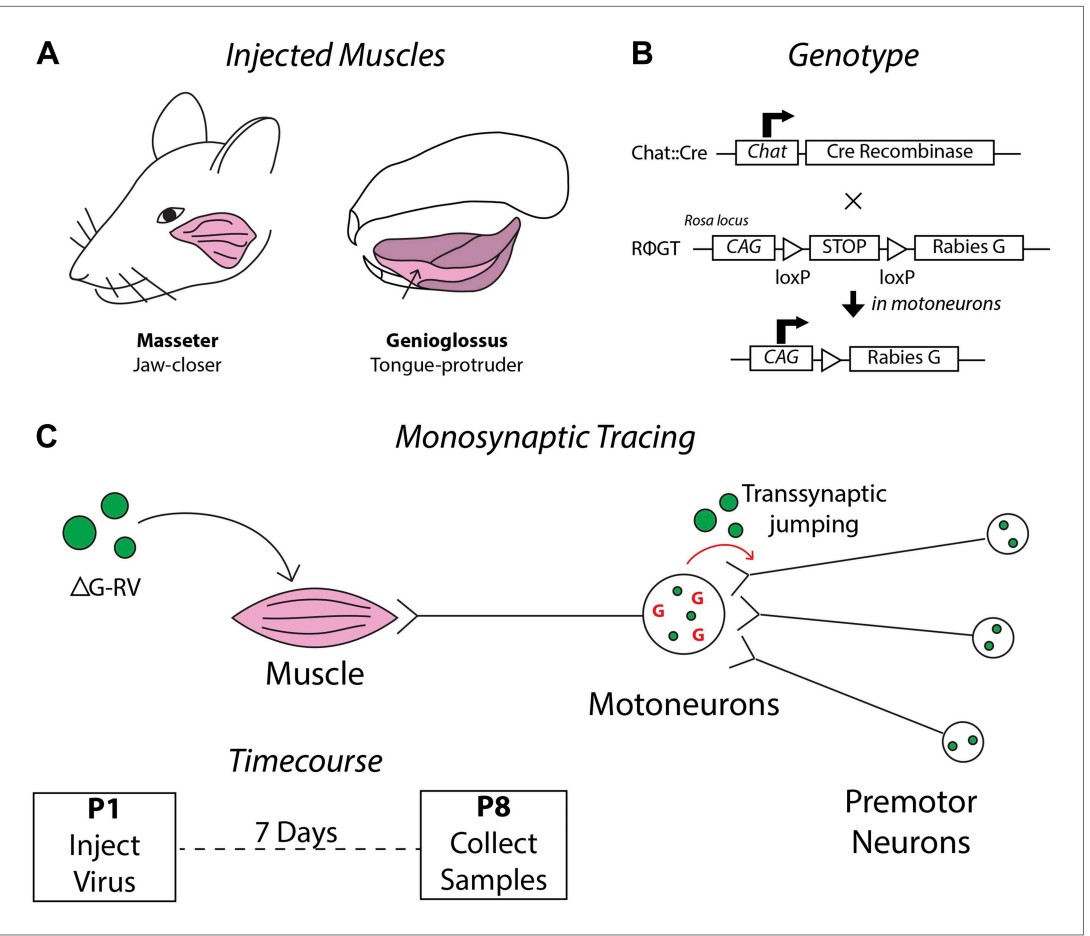

**Figure 1**. Schematics detailing the premotor circuit tracing strategy. (**A**) Illustration of viral injection sites used in this study. *Left*, the jaw-closing masseter muscle; *right*, the genioglossus: a muscle of the tongue controlling protrusion. (**B**) Genetic cross used in this study. Arrow indicates action of Cre recombinase on the RΦGT locus enabling rabies G expression in motoneurons. (**C**) ΔG-RV injection into a selected muscle results in infection of motor axons innervating that muscle. Complementation of the virus with endogenous rabies G in motoneurons results in transsynaptic retrograde labeling of premotor neurons. Retrograde passage is halted at one synapse due to lack of complementation in premotor neurons. *Inset,* pups were injected at post-natal day 1 (P1), and their brainstems were analyzed at post-natal day 8 (P8).
The following figure supplements are available for figure 1:

**Figure supplement 1**. Extremely rare labeling of ChAT+ premotor neurons in masseter and genioglossus premotor tracing studies.

and methods' and associated *Figure 1—figure supplement 1*. The masseter is the primary jaw-closing muscle and is innervated by motoneurons located in the trigeminal motor nucleus (MoV). Masseter activity is coordinated with other muscles in multiple orofacial behaviors including suckling, chewing, biting, and vocalization (*Travers et al., 1997*). To investigate the premotor circuitry control-ling the masseter motoneurons, we injected ΔG-RV-EGFP into the left masseter of P1 Chat::Cre; RΦGT mouse pups (*Figure 1*). 7 days later at P8 (P1→P8 tracing), we serial sectioned and imaged the brains to visualize the transsynaptically labeled masseter premotor neurons.

Video 1 is a representative example of all serial sections from one such transsynaptic tracing experi-ment showing all viral-labeled regions. *Figure 2* shows representative images from selected labeled regions. As a summary of the key findings, the masseter premotor circuitry contains: (1) extensive populations of neurons located bilaterally in the brainstem intermediate reticular (IRt) and lateral retic-ular nuclei (MdRt, PCRt), extending from caudal (*Figure 2A*) to rostral (*Figure 2B, D*) regions; (2) a

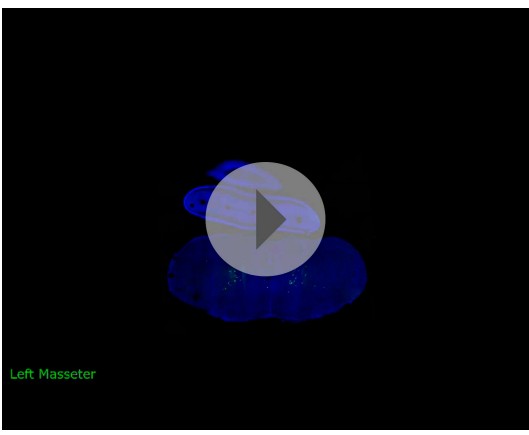

**Video 1**. Representative complete premotor circuit labeling after injection of ΔG-RV-EGFP into the left masseter muscle. Sections were obtained from the brainstem of an 8-day-old pup 7 days after peripheral rabies injection. 80-µm serial sections are shown in sequence from caudal to the hypoglossal motor nucleus (MoXII) to the rostral end of labeling in the dorsal midbrain reticular formation (dMRf).

large number of proprioceptive trigeminal mesencephalic neurons (MesV) (*Figure 2F*), and scattered second-order sensory-related neurons in the rostral and dorsal trigeminal brainstem nuclei (SpO, dPrV, *Figure 2D,E*); (3) numerous neurons in the region surrounding MoV (*Figure 2E*); (4) neurons in deep cerebellar nuclei—in particular the fastigial nucleus (DCN, *Figure 2C*), midbrain reticular formation (dMRf, *Figure 2G*) and the red nucleus (RN, *Figure 2H*); and (5) sparse and sporadically labeled neurons in midline and other regions, including interneurons located in ipsi- and contralateral MoV (*Table 1*). A much more detailed description and quantification of the labeling results for each anatomical location are shown in *Table 1* (n = 5 mice). Note that due to the dense labeling of motoneurons, as well as all axon projections from labeled premotor neurons, we could not accurately quantify numbers of interneurons inside the ipsilateral MoV. Labeling in all other regions is quantified (*Table 1*).

The masseter premotor neurons revealed here through P1→P8 tracing likely reflect the circuits controlling suckling at this early stage, because chewing movements in mice emerge around post-natal day 12 (*Westneat and Hall, 1992*). On the other hand, previous studies have found that during the development of chewing, glycine switches from providing excitatory to inhibitory input onto motoneurons (*Inoue et al., 2007*; *Nakamura et al., 2008*). Thus, it is possible that the same circuitry is used to produce rhythmic suckling early in life and rhythmic chewing later in development (*Langenbach et al., 1992*; *Westneat and Hall, 1992*; *Morquette et al., 2012*).

To investigate whether there might be developmental changes in the masseter premotor circuitry after chewing has begun, we conducted monosynaptic rabies-mediated tracing at P8 and sampled at P15 (P8→P15 tracing) (*Figure 2—figure supplement 1*). A much lower efficiency of motoneuron infection from peripheral injection was observed at this later stage (19 ± 4 motoneurons labeled in P8 injected animals; 35 ± 6 motoneurons labeled in P1 injected animals; mean ± SEM, n = 3 samples per group), and the overall number of labeled premotor neurons was drastically reduced. Despite this, the labeled neurons were distributed in similar locations as those observed in the P1→P8 tracing (*Figure 2—figure supplement 1B–D*). Interestingly, in contrast to other regions, the number of labeled premotor neurons in the lateral paragigantocellular (LPGi) nucleus increased (*Figure 2—figure supplement 1E*). When normalized against the number of infected motoneurons, twice as many LGPi neurons were labeled in the P8→P15 tracing, suggesting more LPGi neurons form synapses with motoneurons, or individual LPGi neurons form synapses with more motoneurons. Due to the overall inefficiency of infection and transsynaptic spreading in P8 or older animals, all subsequent results were obtained from P1→P8 tracing experiments.

## Identifying premotor neurons that directly project to both left and right masseter motoneurons

Because the mandible is joined by ligaments at the midline, jaw movement is obligated to be temporally symmetric on the left and right side. Trigeminal motoneurons do not themselves project bilaterally to enable such coordination (*Shigenaga et al., 1988*). Previously, there have been observations of commissural interneurons located inside MoV projecting to the contralateral MoV (*Appenteng and Girdlestone, 1987*; *Ter Horst et al., 1990*; *Juch et al., 1993*; *McDavid et al., 2006*), raising the possibility that these MoV interneurons might play critical role in left–right symmetry. We also observed labeled interneurons in the contralateral MoV in our tracing, however this labeling was very sparse (see *Figure 3E,F*, arrow heads). Additionally, previous studies using retrograde dyes have labeled some reticular neurons projecting to both the left and right MoV, suggesting that bilateral coordination

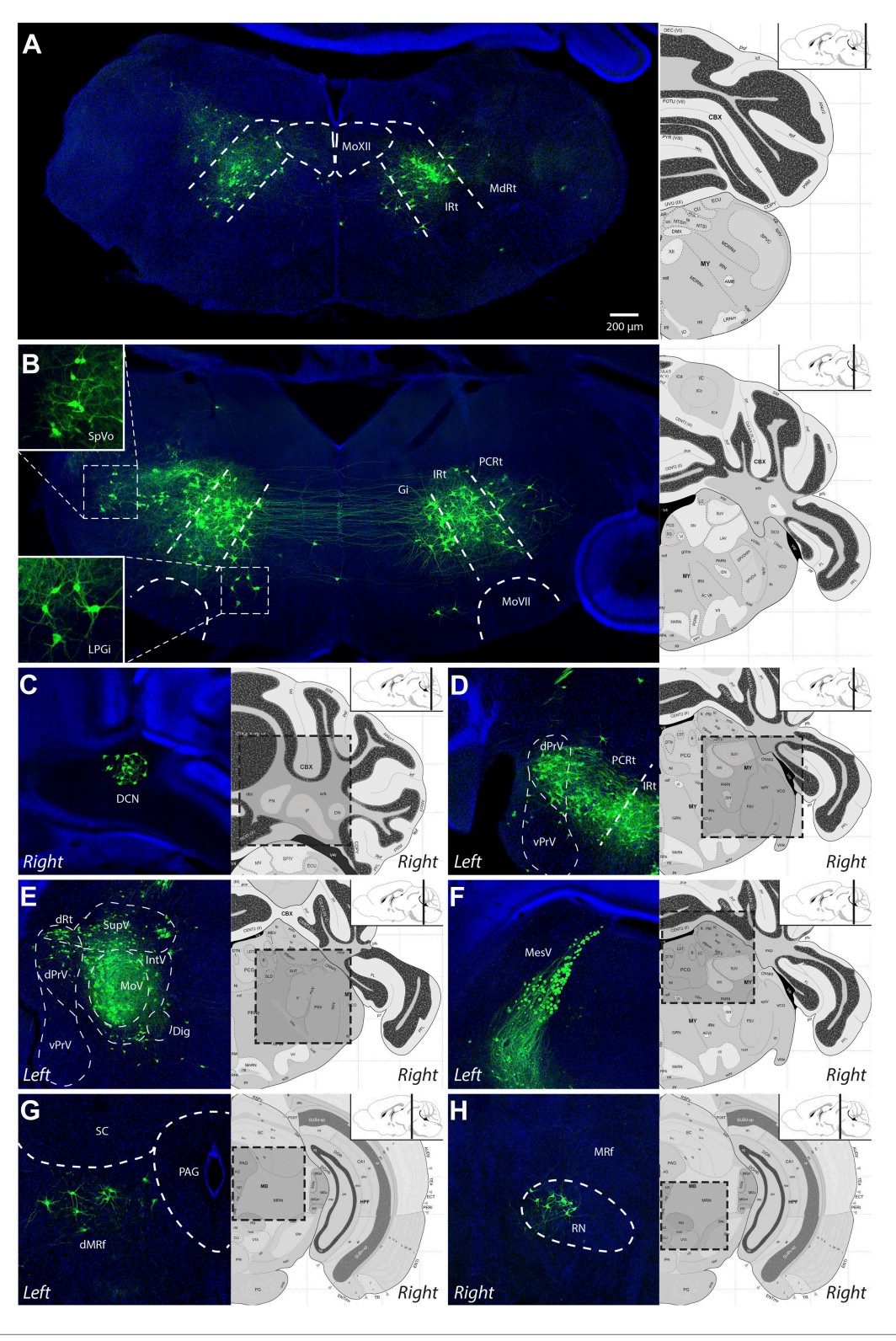

**Figure 2**. Representative images of labeled jaw premotor neurons after ΔG-RV injection into the left masseter muscle. (**A**) Caudal brainstem showed labeling primarily in the caudal intermediate reticular formation (IRt-c) and medullary reticular formation (MdRt). (**B**) Rostral brainstem at the level of the facial motor nucleus (MoVII) showed extensive labeling in the rostral IRt (IRt-r) and some labeling in the gigantocellular (Gi) and parvocellular (PCRt)

*Figure 2. Continued on next page*

*Figure 2. Continued*

reticular formation. *Insets*, labeled neurons in the spinal trigeminal nucleus oralis (SpVo) and the lateral paragigantocellular nucleus (LPGi). This region of the brainstem contained extensive axon collaterals crossing the midline. (**C**–**H**) Labeling of other premotor neuron groups including: the deep cerebellar nuclei (DCN, **C**); the dorsal principal trigeminal sensory nucleus (dPrV, **D**); the motor trigeminal nucleus (MoV, primary infection) and surrounding trigeminal regions (collectively, PeriV) (**E**); the mesencephalic sensory nucleus (MesV), which extended from MoV to dorsal to the periaqueductal grey (PAG) (**F**); the dorsal midbrain reticular formation (dMRf, **G**); and the red nucleus (RN, **H**). Displayed side of the brainstem is indicated in each panel. Stereotaxic maps for this and all subsequent figures were obtained from the Allen Brain Institute website: www.brain-map.org.

The following figure supplements are available for figure 2:

**Figure supplement 1**. The masseter premotor circuit contains more LPGi neurons in old pups.

---

may also arise from inputs other than MoV interneurons (***Kamogawa et al., 1994***; ***Yoshida et al., 2005***). Our finding that masseter motoneurons on one side receive extensive premotor inputs from both ipsi- and contralateral reticular (Rt) neurons (***Figure 2A–B***) suggests that premotor neurons in the reticular formation may be involved in producing synchronized and symmetric jaw motor activity on both sides.

Since the monosynaptic rabies virus expresses fluorescent protein at high levels, axon terminals of viral-infected neurons can be clearly visualized. When premotor neurons were labeled from viral injection into the left masseter muscle, we found that the right MoV nucleus was covered by fluorescently labeled axons (***Video 1***), indicating that many premotor neurons project to motor nuclei on both sides. This is consistent with previous dye tracing studies (***Kamogawa et al., 1994***; ***Yoshida et al., 2005***; ***Kondo et al., 2006***). However, because MoV contains both motoneurons and interneurons (***Appenteng and Girdlestone, 1987***; ***Ter Horst et al., 1990***; ***Juch et al., 1993***; ***McDavid et al., 2006***), it is possible that these bilaterally projecting premotor neurons synapse with motoneurons on the ipsilateral side, and with interneurons on the contralateral side. To determine whether at least some of the contralateral projections directly innervate motoneurons, we used anti-ChAT immunostaining to visualize motoneurons in ΔG-RV-EGFP-labeled brains. Indeed, numerous GFP+ boutons from labeled left masseter premotor neurons directly contacting ChAT+ motoneurons in the right MoV (***Figure 3G,H***).

To further confirm the existence of neurons presynaptic to motoneurons on both sides, as well as to identify the locations of these neurons in the jaw premotor circuitry, we injected the left masseter of P1 Chat::Cre; RΦGT pups with ΔG-RV-EGFP (green), and the right masseter of the same pups with ΔG-RV-mCherry (red). Thus, if any premotor neuron provided monosynaptic input to both the equivalent left and right masseter motoneurons, it would be labeled by both red and green ΔG-RV, thereby appearing yellow. We observed many double-labeled yellow neurons in most of the premotor nuclei, including many in the caudal and rostral reticular formation (***Figure 3A–C***), a few primary proprioceptive neurons in the trigeminal mesencephalic nucleus (MesV) (as distinguished from interneurons by their large spherical unipolar cell bodies (***Lazarov and Chouchkov, 1995***; ***Verdier et al., 2004***)) (***Figure 3D*** and inset), neurons in dorsal principle trigeminal nucleus (dPrV) (***Figure 3C,E,F***), and neurons in the region above dPrV (dRt) (***Figure 3C,F***). The number of double-labeled neurons was relatively few (~8%). However, this is likely a significant under-representation. The masseter is a large muscle, and the chances of virus infecting functionally equivalent muscle fibers on both sides are very low to begin with. Additionally, when taking into account the stochastic nature of viral spreading at the synapses, the actual number of bilateral-projecting premotor neurons could be much higher than what we observed. The presence of premotor neurons innervating motoneurons on both sides throughout the jaw premotor circuitry provides a simple mechanism for directly synchronizing bilateral motoneuron activities.

## Transsynaptic mapping of the premotor circuitry controlling the tongue-protruding hypoglossal motoneurons

Suckling in neonates and chewing later in life involve not only coordination of bilateral jaw muscles, but also coordination between muscles of the jaw, the lips, and the tongue (***Naganuma et al., 2001***; ***Thexton et al., 2004***). We next wanted to map the premotor circuitry controlling the tongue, which is

**Table 1.** Description and quantification of the distribution of masseter premotor neurons

**Masseter premotor neurons**

| Premotor region | % Ipsilateral | % Contralateral |
|---|---|---|
| Reticular regions | | |
| Medullary reticular formation, caudal intermediate reticular formation | 5.65 ± 0.76 | 5.45 ± 0.69 |
| Rostral intermediate reticular formation | 19.41 ± 1.61 | 14.68 ± 1.22 |
| Parvocellular reticular formation | 13.43 ± 0.98 | 5.32 ± 0.32 |
| Lateral paragigantocellular nucleus | 0.44 ± 0.03 | 0.17 ± 0.02 |
| Trigeminal sensory regions | | |
| Mesencephalic sensory nucleus | 16.81 ± 3.98 | 1.11 ± 0.35 |
| Peri-trigeminal zone | 8.16 ± 0.59 | 1.91 ± 0.22 |
| Dorsal principal trigeminal sensory nucleus | 2.47 ± 0.70 | 1.31 ± 0.37 |
| Spinal trigeminal nucleus, Oralis | 1.78 ± 0.21 | 0.31 ± 0.08 |
| Descending control regions | | |
| Dorsal midbrain reticular formation | 0.45 ± 0.21 | 0.08 ± 0.03 |
| Deep cerebellar nuclei | 0.18 ± 0.10 | 0.40 ± 0.06 |
| Red nucleus | 0.01 ± 0.01 | 0.48 ± 0.12 |

Extensive bilateral labeling in both caudal (level of MoXII; MdRt, IRt-c) and rostral (rostral to MoXII to caudal MoV; IRt-r, PCRt) reticular regions was observed. Trigeminal sensory-related nuclei labeling primarily included MesV, comprised of jaw muscle proprioceptive and periodontal sensory neurons, and rostral trigeminal sensory nuclei (SpVo, dPrV, and PeriV). Labeling in MesV and SpVo showed a strong ipsilateral bias. Nuclei implicated in descending control were labeled, consisting of contralateral DCN and RN, and ipsilateral dMRf, as well as LPGi. We also found scattered and sparse labeling of premotor neurons in the Gi, interneuron labeling in the contralateral MoV, lateral reticular formation, pre-Bötzinger complex (pre-BötC), medial vestibular nucleus, raphe magnus nucleus, raphe pallidus nucleus, dorsal medial tegmental nucleus, and pontine reticular nucleus. However the labeling pattern and number of neurons in these nuclei were few and not consistent across animals. Percentage of total premotor neurons in a region was calculated within sample (thereby normalizing values to tracing efficacy), and subsequent values were averaged across five mice. All values are averages ± SEM.

innervated by motoneurons located in the hypoglossal motor nucleus (MoXII). The genioglossus is the main tongue-protruding muscle. Partly due to its ease of access, the genioglossus has been a primary target for many previous studies investigating orofacial behaviors, including chewing, suckling, and licking (*Gerstner and Goldberg, 1991*; *Sawczuk and Mosier, 2001*; *Kakizaki et al., 2002*). During rhythmic chewing, the genioglossus and masseter are activated with the same rhythm but in opposite phases, because activation of both at once could result in biting of the tongue. We investigated the premotor circuitry of the genioglossus motoneurons using a similar method as described for the masseter motoneurons (*Figure 1*). ΔG-RV-EGFP was injected into the left genioglossus muscle of the tongue. However, due to the small size of the tongue in P1 mice and the amount of virus necessary to infect motor axons, in most animals some of the virus injected to the left side unavoidably spread to the right genioglossus. Thus, it was not feasible to map the premotor inputs solely to the genioglossus on one side.

*Video 2* is a representative example of one serially sectioned brain after ΔG-RV-EGFP-mediated transsynaptic tracing from primarily the left genioglossus muscle. *Figure 4* shows representative labeling patterns from selected brainstem regions. As a summary of the key findings, the genioglossus premotor circuitry contains: (1) a large population of premotor neurons located bilaterally in the intermediate reticular formation (IRt), with fewer neurons in the lateral side of the reticular formation (in PCRt) (*Figure 4A–C*); (2) premotor neurons located in trigeminal sensory nuclei mainly in caudal brainstem (SpC, *Figure 4A*), and a few in dPrV (*Figure 4E*) and MesV (*Figure 4F*), as well as taste-related neurons in the nucleus of the solitary tract (NTS, *Figure 4A–B*); (3) neurons in the deep cerebellar nucleus (DCN, *Figure 4D*) and midbrain reticular formation (dMRf, *Figure 4G*), which likely provide descending inputs; and (4) scattered cells in the midline and other brainstem structures (*Table 2*).

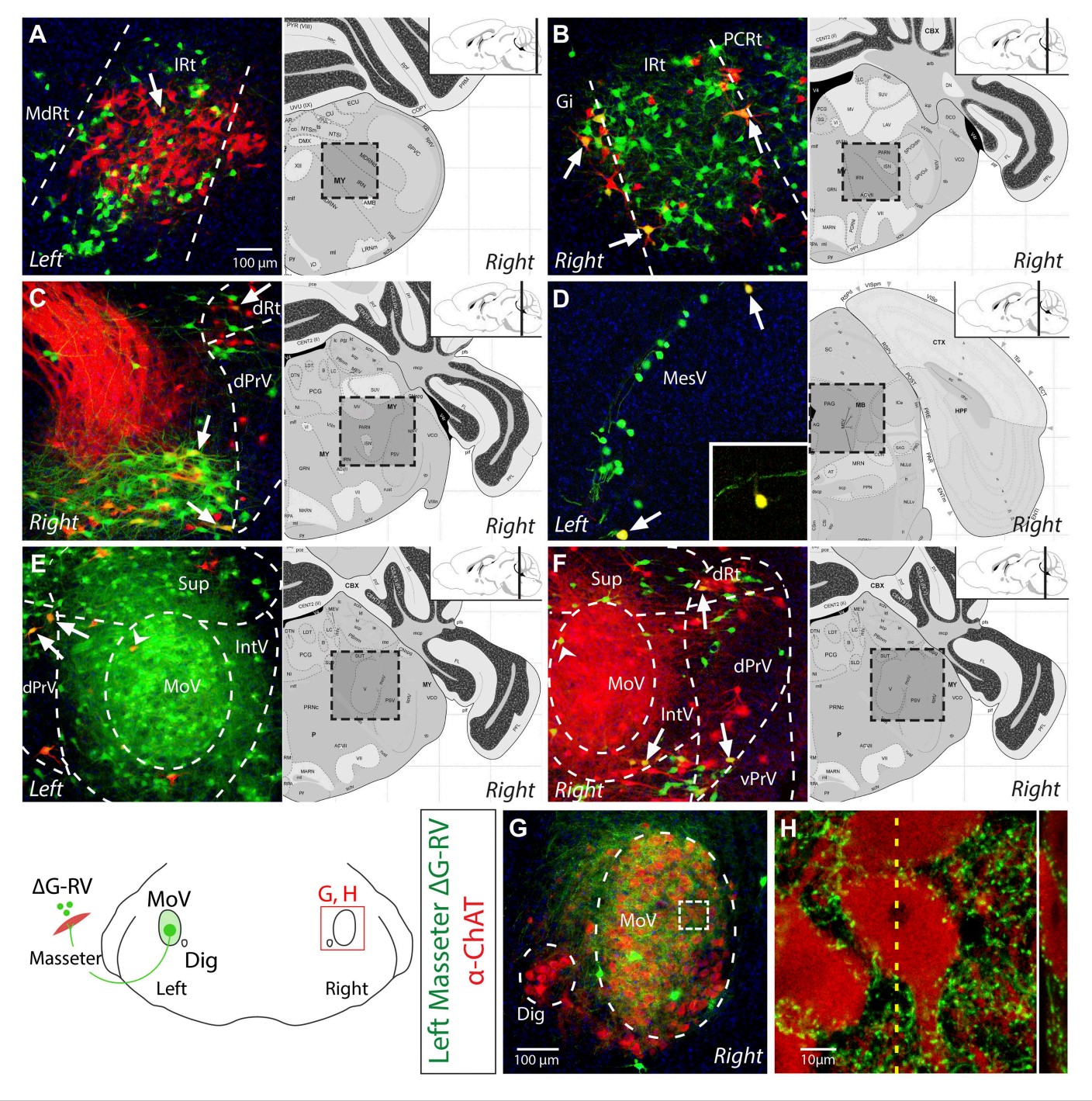

**Figure 3**. Evidence for the presence of bilateral-projecting masseter premotor neurons. (A–F) Simultaneous tracing of left (ΔG-RV-EGFP, green) and right (ΔG-RV-mCherry, red) masseter premotor neurons. Yellow cells, which indicate bilaterally projecting premotor neurons, were observed in many brainstem regions (arrows indicate some examples) including: IRt-c (**A**); IRt-r and Gi (**B**); PCRt and the dorsal reticular region (dRt) (**C**); MesV, with a magnified (1.5X) view of the upper double-labeled neuron highlighting its morphology characteristic of primary afferent neurons in MesV (**D**); and dPrV and the peri-trigeminal region (PeriV) (**E** and **F**). Additionally, premotor interneurons were found in the contralateral MoV (arrow heads **E** and **F**). Displayed side of the brainstem is indicated in each panel. (**G–H**) ChAT-immunostained (red) contralateral MoV showing extensive innervation from labeled ipsilateral masseter premotor axons (green). The boxed region in **G**, and a line scan (*right*) of an orthogonal slice at the yellow dotted line are shown in **H**.

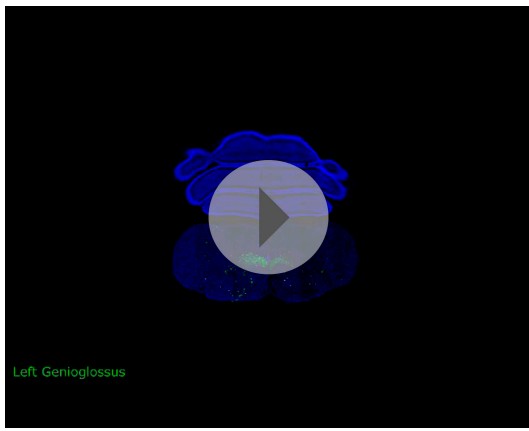

Left Genioglossus

**Video 2**. Representative complete premotor circuit labeling after injection of ΔG-RV-EGFP into the genioglossus muscle. Sections were obtained from the brainstem of an 8-day-old pup 7 days after peripheral rabies injection. 80-µm serial sections are shown in sequence from caudal to the hypoglossal motor nucleus (MoXII) to the rostral end of labeling in the dorsal midbrain reticular formation (dMRf).

A much more detailed description and quantification of the labeling results for each anatomical location are shown in *Table 2* (n = 5 mice).

## Genioglossus premotor neurons also provide inputs to specific facial and jaw-opening motoneurons

Very interestingly, from the genioglossus trans-synaptic tracing experiments, we observed that a large cohort of axon collaterals from these tongue-protruder premotor neurons projected to the motoneurons innervating the jaw-opening digastric muscle (anterior portion), which are located in the accessory trigeminal motor nucleus (Dig) medial and ventral to the main MoV (*Figure 4E*, arrow). Furthermore, many genioglossus premotor axons were observed projecting into the central MoVII, but not other divisions of MoVII, on both sides of the brainstem (*Figure 4C*, arrow; *Figure 5C*). It is known that motoneurons in this central MoVII supply the posterior digastric muscle (jaw-opening and swallowing), the platysma muscle (jaw-depressing and lip-lowering), and the lower lip muscle (*Ashwell, 1982*; *Hinrichsen and Watson, 1984*). To further examine whether these collateral projections indeed form synapses onto Dig or central MoVII motoneurons, we conducted anti-ChAT immunostaining on brainstems after ΔG-RV-EGFP tracing from the genioglossus. Using high-resolution confocal microscopy, we observed GFP+ boutons directly contacting ChAT+ motoneurons in both the central MoVII (*Figure 5C,D*) and the Dig (*Figure 5A,B*). These findings confirm that subsets of genioglossus premotor neurons simultaneously provide inputs to digastric and/or platysma/lower lip motoneurons. Considering many previous experimental observations of the co-activation of these muscles (*Gerstner and Goldberg, 1991*; *Liu et al., 1993*; *Travers et al., 1997*; *Thexton et al., 1998*; *Naganuma et al., 2001*), for example tongue protrusion always involves concomitant opening of the jaw and movement of the lower lip, such specifically shared premotor neurons are the simplest mechanism to generate co-activation of their target motoneurons.

## Masseter premotor neurons also provide inputs to tongue-retractor motoneurons in dorsal MoXII

After observing genioglossus premotor neurons also innervating lip-lowering and jaw-opening motoneurons, we wondered if this type of multi-target connectivity was also employed by masseter premotor neurons. Previous studies using retrograde tracers discovered the existence of neurons which project to both MoV and MoXII nuclei (*Amri et al., 1990*; *Li et al., 1993*; *Travers et al., 2005*). The masseter exhibits some degree of synchronous activity with tongue-retractor muscles during feeding behavior (*Kakizaki et al., 2002*), although they are not as tightly coordinated as the digastric and genioglossus muscles. Tongue-retractor motoneurons are located in the dorsal MoXII, as compared to tongue-protruder genioglossus motoneurons which are located in the ventral MoXII (*McClung and Goldberg, 1999*). We thus re-examined the MoXII in ChAT-immunostained sections after ΔG-RV infection of the left masseter muscle (*Figure 5E–H*). We found that indeed a few masseter premotor axons specifically innervated dorsal MoXII (*Figure 5E,G*), and formed boutons opposing both contralateral and ipsilateral motoneurons located in this region (*Figure 5F,H*). Thus, shared premotor neurons innervating both jaw-closing and some tongue-retracting motoneurons could indeed facilitate the co-activation of these two muscles.

## Distinct and common premotor loci for masseter and genioglossus motoneurons

From the circuit-tracing results described above, it is immediately apparent that masseter and genioglossus motoneurons receive distinct sensory-related inputs. The masseter neurons receive extensive

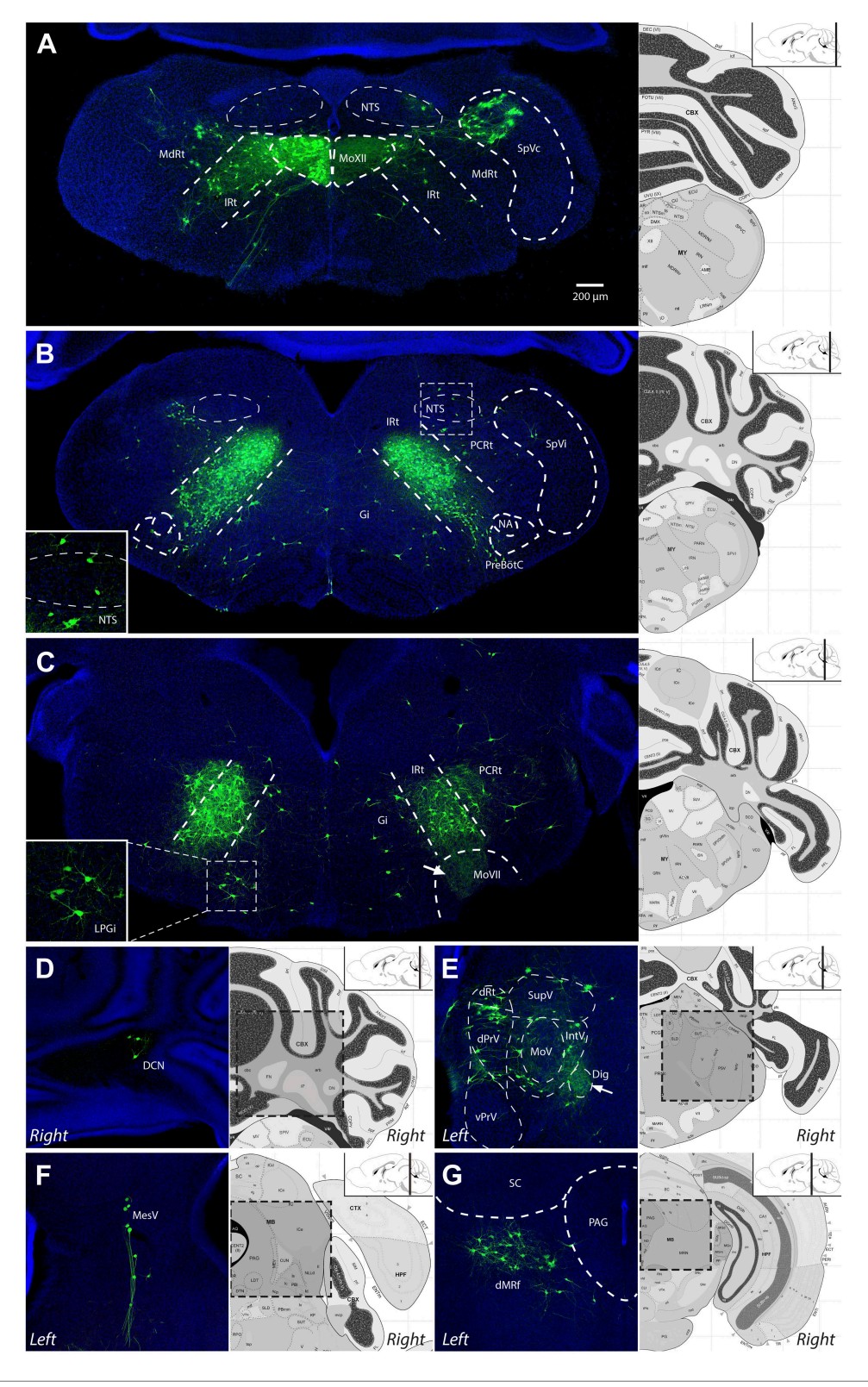

**Figure 4**. Representative images of labeled tongue premotor neurons after ΔG-RV injection into the left genioglossus muscle. (**A**) Caudal brainstem at the level of the hypoglossal motor nucleus (MoXII), illustrating primary infection of the left MoXII, and extensive labeling in IRt-c, in the spinal trigeminal nucleus caudalis (SpVc), and sparse
*Figure 4. Continued on next page*

*Figure 4. Continued*

labeling in the nucleus of the solitary tract (NTS). (**B**) Brainstem between MoXII and MoVII showing extensive bilateral labeling in the IRt-r, with sparse labeling in the NTS, PCRt, Gi, and pre-Bötzinger complex (pre-BötC) below the nucleus ambiguus (NA). (**C**) Rostral brainstem at the level of MoVII, showing labeling in the IRt-r and extending into the PCRt. Additionally, bilateral labeling of the LPGi is visible (*inset*). Labeled premotor axon collaterals are visible invading the central MoVII (arrow). (**D–G**) Other groups of labeled premotor neurons in: the DCN (**D**); dPrV, dRt and PeriV (**E**); MesV (**F**); and dMRf (**G**). Genioglossus premotor labeling was weaker in the DCN and stronger in bilateral dMRf and LPGi as compared to masseter premotor labeling. Note the dense innervation of the anterior digastric motor nucleus (Dig) with premotor axons (**E**, arrow). Displayed side of the brainstem is indicated in each panel.

**Table 2.** Description and quantification of the distribution of genioglossus premotor neurons

**Genioglossus premotor neurons**

| Premotor region | % Ipsilateral | % Contralateral |
| --- | --- | --- |
| Reticular regions | | |
| Caudal intermediate reticular formation | 18.84 ± 1.79 | 8.99 ± 0.78 |
| Rostral intermediate reticular formation, parvocellular reticular formation | 27.85 ± 0.45 | 24.23 ± 1.78 |
| Lateral paragigantocellular nucleus | 0.85 ± 0.23 | 0.73 ± 0.18 |
| Trigeminal sensory regions | | |
| Spinal trigeminal sensory nucleus, caudalis | 4.21 ± 1.44 | 2.19 ± 0.70 |
| Peri-trigeminal zone | 2.66 ± 0.47 | 1.88 ± 0.30 |
| Mesencephalic sensory nucleus | 1.53 ± 0.31 | 0.59 ± 0.10 |
| Spinal trigeminal sensory nucleus, oralis | 0.95 ± 0.20 | 0.56 ± 0.06 |
| Dorsal principal trigeminal sensory nucleus | 0.73 ± 0.25 | 0.52 ± 0.13 |
| Descending control regions | | |
| Dorsal midbrain reticular formation | 1.18 ± 0.38 | 1.21 ± 0.33 |
| Deep cerebellar nuclei | 0.11 ± 0.03 | 0.08 ± 0.02 |
| Red nucleus | 0.06 ± 0.04 | 0.05 ± 0.04 |

Extensive bilateral labeling was observed in a concentrated band within the IRt from the medulla to the caudal border of MoVII (IRt-c, IRt-r), after which it spread slightly into the PCRt (IRt-r, PCRt). Labeling in trigeminal sensory related nuclei was primarily in the caudal sensory nuclei, particularly in bilateral SpVc. Additional sparse labeling of neurons in trigeminal sensory-related regions was found in SpVi, dPrV, PeriV, and MesV, with the MesV labeling occurring as far rostral as dorsal to the PAG. Nuclei implicated in descending control were labeled, consisting of contralateral DCN, bilateral dMRf, and bilateral LPGi. We also found scattered and sparse labeling of premotor neurons in the Gi, nucleus of the solitary tract (NTS), rostral ventral respiratory group, lateral reticular nucleus, pre-BötC, midline raphe nuclei, superior vestibular nucleus, pontine reticular nucleus, and dorsal medial tegmental area. However, the labeling pattern and number of neurons in these nuclei were few and not consistent across animals. Percentage of total premotor neurons in a region was calculated within sample (thereby normalizing values to tracing efficacy), and subsequent values were averaged across five samples. All values are averages ±SEM.

MesV-derived proprioceptive and SpVo-derived somatosensory inputs, and the genioglossus neurons primarily receive SpVc-derived somatosensory inputs (compare ***Figure 2D,F*** with ***Figure 4A,F***), while both motoneuron populations receive sensory-related input from dPrV. Additionally, genioglossus motoneurons receive taste-related inputs from the NTS (***Figure 4B***, inset), while no labeling in the NTS was observed in the masseter premotor circuit. However, in the rostral reticular formation where many of the premotor neurons for both muscles are located, these populations are distributed in a similar pattern, suggestive of spatial intermingling of these two groups of premotor neurons.

To more directly compare the spatial distributions of jaw and tongue premotor inputs, we injected ΔG-RV-EGFP (green) into the left genioglossus and ΔG-RV-mCherry (red) into the left masseter muscle

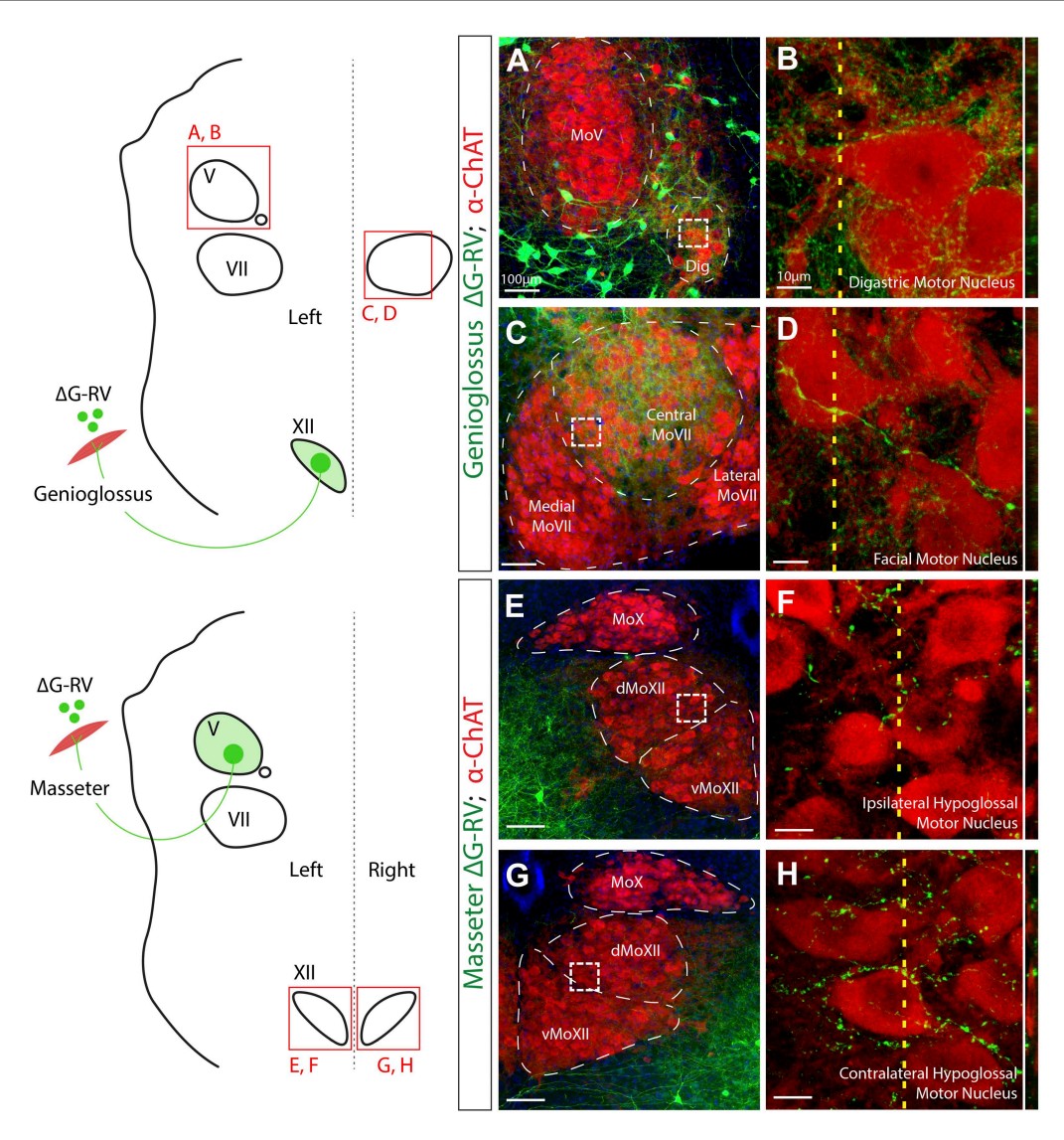

**Figure 5**. Premotor axon boutons onto ChAT+ motoneurons revealing direct premotor control of multiple motor groups. (**A**) ChAT-immunostained (red) jaw-opening digastric (Dig) motoneurons showing innervation from labeled genioglossus premotor axons (green). (**B**) The boxed region in **A**, and a line scan (*right*) of an orthogonal slice at the yellow dotted line in **A**. (**C**) ChAT-immunostained central MoVII showing innervation from labeled genioglossus premotor axons. (**D**) The boxed region in **C**, and a line scan (*right*) of an orthogonal slice at the yellow dotted line in **C**. (**E**) ChAT-immunostained left MoXII showing innervation from labeled left masseter premotor axons. (**F**) The boxed region in **E**, and a line scan (*right*) of an orthogonal slice at the yellow dotted line in **E**. (**G**) ChAT-immunostained right MoXII showing innervation from labeled left masseter premotor axons. (**H**) The boxed region in **G**, and a line scan (*right*) of an orthogonal slice at the yellow dotted line in **G**.

of Chat::Cre; RΦGT pups (*Figure 6*; *Video 3*; *Video 4*). We focus our comparison on spatial distribution rather than the absolute numbers of labeled-neurons due to varying levels of infection in the two muscles across different samples.

In the caudal-most IRt at the level of MoXII, masseter premotor neurons are situated ventrally to the labeled genioglossus premotor neurons (*Figure 6A*). At the level just rostral to the MoXII, sparsely labeled masseter premotor neurons are spatially intermixed with densely labeled genioglossus premotor neurons in the IRt (*Figure 6B*). At the level of the MoVII and MoV, there is extensive spatial overlap between the two populations of premotor neurons in the reticular region (IRt and PCRt,

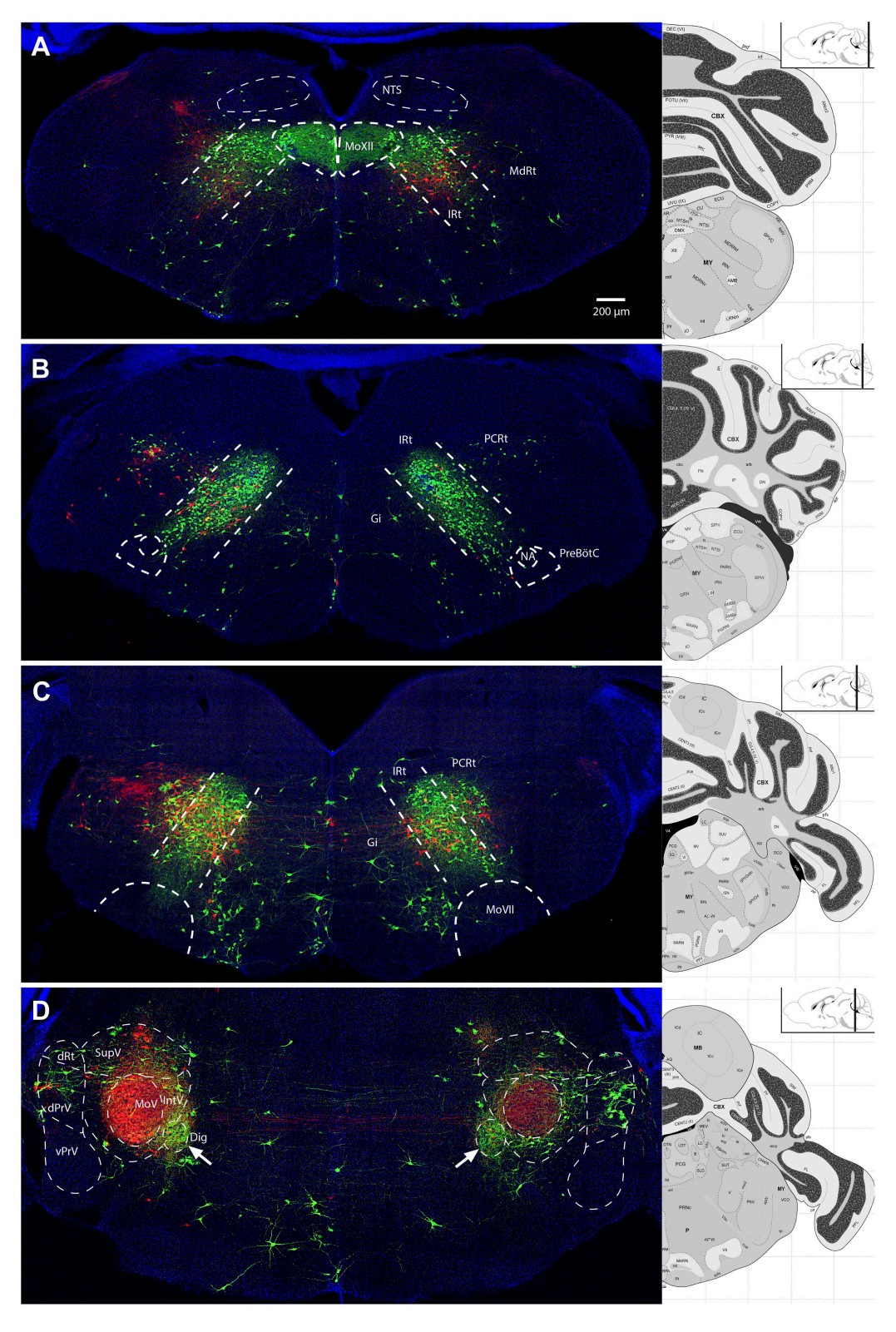

**Figure 6**. Representative images from experiments simultaneously tracing both genioglossus (green) and left masseter (red) premotor neurons. (**A**) Caudal brainstem at the level of MoXII, showing a rough spatial segregation between the two premotor populations in the IRt-c, with masseter premotor neurons more ventrally situated as compared to genioglossus premotor neurons. (**B**) Brainstem at the level between MoXII and MoVII, showing mostly genioglossus premotor neurons present in this region. (**C**) Brainstem at the level of MoVII, showing spatial intermingling of the two premotor

*Figure 6. Continued on next page*

*Figure 6. Continued*

populations. Axon collaterals crossing the midline are visible from both genioglossus and masseter premotor neurons. Genioglossus axon collaterals are visible extending into the central MoVII. (**D**) Labeling patterns at the level of MoV in dPrV, dRt (region just above dPrV), supra trigeminal nucleus (SupV), and inter-trigeminal region (IntV). Masseter premotor axon collaterals (red) extend into the contralateral MoV, while genioglossus premotor axon collaterals (green) extend into the Dig (arrows). Displayed side of the brainstem is indicated in each panel.

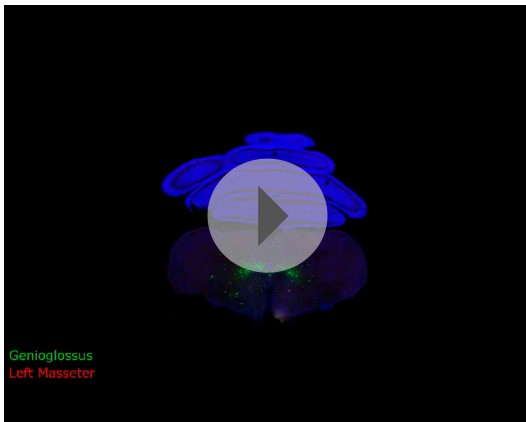

**Video 3**. Comparison of masseter (ΔG-RV-mCherry) and genioglossus (ΔG-RV-EGFP) premotor circuitry. Sections were obtained from the brainstem of an 8-day-old pup 7 days after peripheral rabies injection. 80-µm serial sections are shown in sequence from caudal to the hypoglossal motor nucleus (MoXII) to the rostral end of labeling in the dorsal midbrain reticular formation (dMRf).

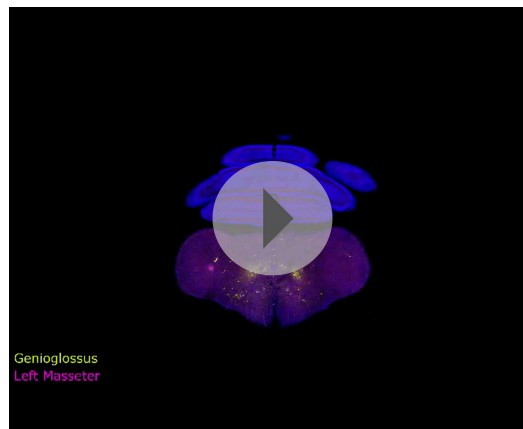

**Video 4**. Pseudocolored movie of masseter (ΔG-RV-mCherry) and genioglossus (ΔG-RV-EGFP) premotor circuitry. Sections were obtained from the brainstem of an 8-day-old pup 7 days after peripheral rabies injection. 80-µm serial sections are shown in sequence from caudal to the hypoglossal motor nucleus (MoXII) to the rostral end of labeling in the dorsal midbrain reticular formation (dMRf). Images are pseudocolored such that masseter infection is visible in magenta, and genioglossus infection is visible in yellow.

*Figure 6C*), and in regions surrounding MoV (*Figure 6D*). Notably, these regions in the reticular formation have previously been found to contain central pattern generating circuitry for jaw and tongue movements (*Chandler and Tal, 1986*; *Lund, 1991*; *Nakamura and Katakura, 1995*; *Morquette et al., 2012*). However, even when the premotor pools were intermixed, we never observed double-labeled neurons; they remain distinct populations, avoiding simultaneous activation of these two muscles which could result in biting of one's own tongue.

## Neurotransmitter phenotypes of jaw and tongue premotor neurons in the brainstem reticular nuclei

The majority of the masseter and the genioglossus premotor neurons are located in the intermediate reticular region (IRt) between the level of MoXII and the rostral end of MoV. Because the rostral portion of this region between the inferior olive and MoV is thought to contain CPGs for jaw and tongue movements (*Chandler and Tal, 1986*; *Lund, 1991*; *Nakamura and Katakura, 1995*; *Morquette et al., 2012*), we wanted to determine the potential output signs of labeled premotor neurons. To do this, we conducted a series of in situ hybridization experiments to examine the neurotransmitter phenotypes while simultaneously immunostaining for EGFP expressed in ΔG-RV-EGFP-labeled premotor neurons. We focused our analysis on known markers for glutamatergic neurons (vesicular glutamate transporter 2: vGluT2), GABAergic neurons (glutamic acid decarboxylase 1 and 2: GAD1 and GAD2 mixed probe [GAD1/2]), and glycinergic neurons (glycine transporter 2: GlyT2). We additionally tested neurons for tryptophan hydroxylase 2 (Tph2) expression as a marker for serotonergic neurons, since we found some premotor neurons in the midline Raphé which is a source of serotonin in the brain.

We performed these in situ-immuno analyses on brains 7 days post ΔG-RV injection. We found that both the masseter and genioglossus premotor neurons in IRt are of mixed transmitter phenotypes, that is they can be either excitatory (vGluT2+), or inhibitory (GAD1/2+ or GlyT2+) (*Figure 7A–H*). These mixed phenotypes are also

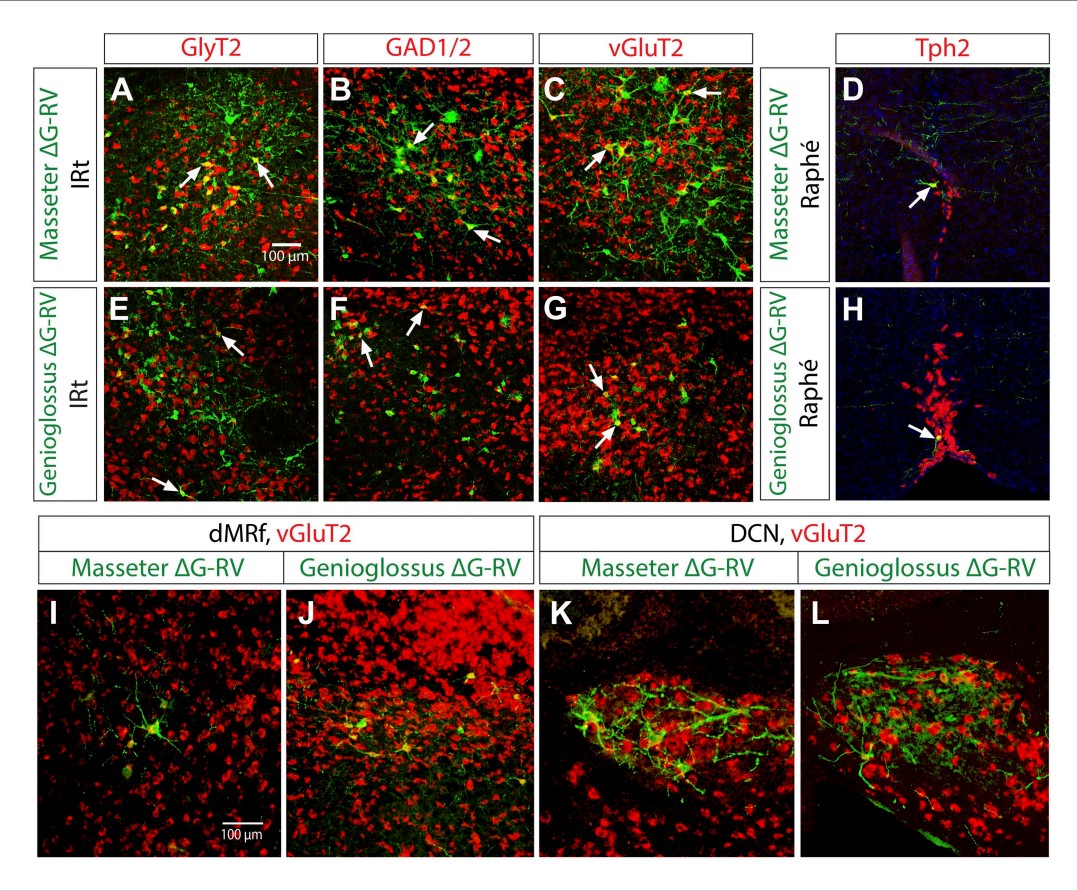

**Figure 7**. Neurotransmitter phenotypes of labeled premotor neurons. In situ hybridization in combination with rabies tracing showing glycinergic (**A** and **E**) GABAergic (**B** and **F**), and glutamatergic (**C** and **G**) premotor neurons to masseter and genioglossus motoneurons. Serotinergic neurons were found in the midline Raphé in both premotor tracing studies (**D** and **H**). Premotor neurons observed in descending regions, including the dMRf (**I** and **J**) and DCN (**K** and **L**) were glutamatergic.

true for premotor neurons in most other regions in brainstem (data not shown). Due to viral toxicity, a portion of GFP+ neurons in these regions did not hybridize with any markers. Therefore, we could not reliably quantify the relative percentage of each type of neuron. However, in all samples examined, we only detected vGluT2+ neurons in the cerebellar DCN and the midbrain dMRf (*Figure 7I–L*), with a notable absence of GAD1/2+ or GlyT2+ neurons in these regions (data not shown) suggesting that the descending inputs from cerebellum and midbrain to jaw and tongue motoneurons are primarily excitatory.

## Discussion

In this study, we used a monosynaptic rabies virus-based tracing technology to systematically map the premotor circuitry for jaw-closing (masseter) and tongue-protruding (genioglossus) muscles. The premotor wiring diagrams for these two motoneuron groups are summarized in *Figure 8D*. Our results provide the anatomical framework for future functional dissection of the neural control of orofacial behaviors. Moreover, we found that in both cases, some premotor neurons traced from one muscle also form synaptic boutons onto distinct motoneurons located in different motor nuclei. These results uncover a set of simple premotor circuit configurations that are ideally suited to orchestrate the orofacial coordination observed in previous behavioral studies as discussed in detail below.

### Implications for bilateral coordination of jaw muscles

Because the mammalian mandible is connected by ligaments at the midline, the output of left and right jaw motoneurons must be temporally symmetric. Even when humans chew more on one side, the

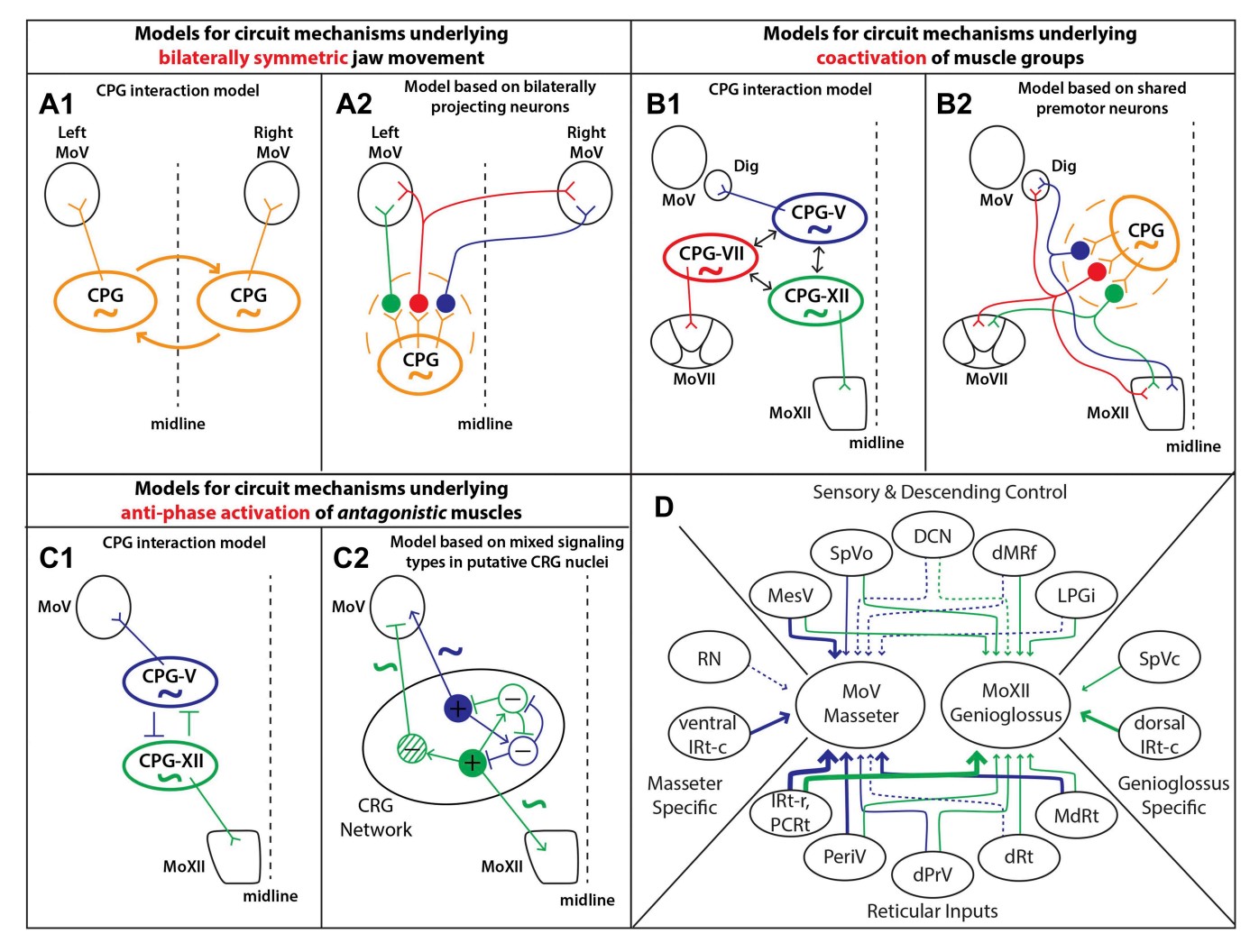

**Figure 8**. Summary of premotor circuit tracing results and models for circuit mechanisms underlying several aspects of motor coordination. (**A**) Models for ensuring bilaterally symmetric jaw movements. (**A1**) Previous model based on interactions of independent CPGs on each side of the brainstem. (**A2**) A model based on this study; signals from one CPG are relayed to motoneurons on both sides through premotor neurons. Dashed circle indicates the possibility of jaw premotor neurons being part of the CPG or distinct targets of the CPG. (**B**) Models for inter-muscular co-activation. (**B1**) Previous model based on interactions between muscle-specific premotor CPGs. (**B2**) A model based on this study: inter-muscular co-activation could occur through a subpopulation of shared premotor neurons that provide the same input to multiple motor groups. (**C**) Models for anti-phase activation of antagonistic muscles. (**C1**) A CPG-interaction model based on reciprocal inhibition of muscle-specific CPGs. (**C2**) A hypothetical model extending from the in situ results and intracellular recordings conducted in previous studies. An interconnected rhythm-generating network could provide both excitatory and inhibitory outputs onto motoneurons. Blue neurons fire to activate masseter motoneurons and inhibit excitatory output onto genioglossus motoneurons (blue rhythmic activity trace), while green neurons fire to activate genioglossus motoneurons and inhibit masseter motoneurons (anti-phase green rhythmic activity trace). (**D**) A summary diagram of the various shared and distinct premotor nuclei containing neurons projecting to the masseter or genioglossus motoneurons. Green represents inputs to genioglossus motoneurons, while blue represents inputs to masseter motoneurons. Thickness of line denotes percentage of premotor inputs arising from a specific nucleus as determined in *Tables 1 and 2*. Large solid line: >30%; medium solid line: 10–30%; small solid line: 1–10%; dotted line: <1%.

contralateral masseter is simultaneously activated to a comparable extent (*Moore, 1993*; *Peyron et al., 2002*). By analogy to the studies investigating spinal locomotion circuits, one model of bilateral coordination involves the interaction between the jaw CPGs in the left and right brainstem (*Figure 8A1*). Studies investigating the rhythm-generating properties of the brainstem after midline transection have supported this theory through the finding that sufficient circuitry exists on both sides of the brainstem to independently generate a rhythmic output (*Chandler and Tal, 1986*; *Ihara et al., 2013*).

Previous studies have implicated commissural interneurons located inside MoV and bilaterally projecting neurons extending axons to both left and right MoV in some brainstem regions as a potential source of bilaterally coordinated inputs (*Appenteng and Girdlestone, 1987*; *Ter Horst et al., 1990*; *Juch et al., 1993*; *Kamogawa et al., 1994*; *Yoshida et al., 2005*; *McDavid et al., 2006*). We also labeled a sparse number of interneurons in the contralateral MoV (*Figure 3E,F*). Furthermore, we observed axon terminals of premotor neurons transsynaptically labeled from left masseter motoneurons extensively innervating the contralateral (right) MoV (*Figure 6C*) and forming boutons opposing motoneurons (*Figure 3G,H*), supporting the existence of large numbers of bilaterally projecting premotor neurons. Using two-color rabies-mediated tracing, we confirmed that premotor neurons that synapse onto left and right masseter motoneurons were present in many brainstem regions. It thus appears that the jaw premotor circuit includes the simplest configuration for bilateral coordination: a single premotor neuron that synapses on equivalent ipsilateral and contralateral motoneurons.

Interestingly, these neurons are especially prevalent in the reticular formation region previously identified as containing the jaw CPG (*Chandler and Tal, 1986*; *Chandler et al., 1990*; *Lund, 1991*; *Nakamura and Katakura, 1995*; *Morquette et al., 2012*). This organization could facilitate bilateral synchrony and symmetry, either independently of or in concert with interactions between CPGs, by enabling the output of the jaw CPG on one side to activate jaw muscles on both sides (*Figure 8A2*). In this model, the premotor neurons that transmit the CPG signals can be either part of the CPG (dashed circle in model) or immediately downstream of the CPG, and can include neurons projecting either unilaterally or bilaterally to the MoV (*Figure 8A2*).

## Implications for coordination of co-activated muscles

Tongue activity is tightly coordinated with jaw and facial muscle activity during feeding behaviors in a variety of mammals, including humans (*Westneat and Hall, 1992*; *Takada et al., 1996*; *Ishiwata et al., 2000*; *Yamamura et al., 2002*). More specifically, the tongue-protruding genioglossus is active in phase with the jaw-opening digastric muscle and with the orbicularis oris of the lips under a wide range of conditions (*Liu et al., 1993*; *Takada et al., 1996*), and the tongue-retracting styloglossus is often active in phase with the jaw-closing masseter muscle (*Kakizaki et al., 2002*). Previous work studying body muscle coordination in vertebrates and invertebrates showed that distinct CPGs which control motoneurons of different body segments may interact with each other to effect cross-muscle coordination (*Cang and Friesen, 2002*; *Briggman and Kristan, 2008*; *Smarandache-Wellmann et al., 2014*). Because orofacial muscles are innervated by different groups of motoneurons located in MoXII, MoV, or MoVII, respectively, an analogous model would be that orofacial muscle co-activation is achieved by interaction between different CPGs that drive distinct motoneuron pools (*Figure 8B1*). Our study discovered that axon collaterals from labeled genioglossus premotor neurons also innervate MoV motoneurons supplying the jaw-opening anterior digastric muscles, as well as the motoneurons located in the central part of the MoVII. The central MoVII supplies the posterior digastric muscle (jaw-opening and swallowing), the platysma muscle (which depresses the jaw and draws down the lower lip), and the lower lip muscle (*Munro, 1974*; *Ashwell, 1982*; *Hinrichsen and Watson, 1984*), which are all activated when the tongue is protruding (*Figure 5*). Additionally, we found that axon collaterals from labeled masseter premotor neurons also innervate motoneurons located in the dorsal MoXII, previously shown to innervate tongue-retractor muscles (*McClung and Goldberg, 1999*). Our results indicate that specific premotor neurons simultaneously innervating multiple different groups of motoneurons provides the simplest circuit mechanism for enabling orofacial muscle co-activation (*Figure 8B2*). It is likely that such common premotor neurons are located in IRt because the majority of labeled jaw or tongue premotor neurons are located in that region.

## Implications for anti-phase coordination of antagonistic muscles

During chewing, the tongue-protruding genioglossus and jaw-closing masseter are generally active with the same rhythm but in opposite phases to prevent biting of the tongue. Again, this could in theory be achieved by mutual inhibition of their separate CPGs, which might have similar rhythm-generation properties (*Figure 8C1*). Here, we found that many premotor neurons for these two antagonist motor groups are located in a spatially intermixed pattern (especially those situated between the caudal end of MoVII and the rostral end of MoV) (*Figure 6C*). As this region is within the region proposed to contain the CPG for chewing (*Chandler and Tal, 1986*; *Nozaki et al., 1986*; *Ihara et al., 2013*), our findings suggest that premotor neurons for different orofacial muscles are embedded

within the same rhythm-generating network. Furthermore, previous studies examining intracellular potentials have found that masseter motoneurons receive rhythmic inhibitory and excitatory signals (**Goldberg et al., 1982**), while genioglossus motoneurons receive only rhythmic excitatory signals (**Sahara et al., 1988**). In the reticular region previously implicated in rhythm generation, we found that premotor neurons consisted of both excitatory and inhibitory neurons. Thus, we propose a hypothetical model consistent with these results whereby premotor neurons in an extended rhythmogenic network could have the same rhythm but could have opposite phases of activity through local circuit interactions, resulting in rhythmic excitation of masseter motoneurons and anti-phase excitation of genioglossus motoneurons and inhibition of masseter motoneurons (**Figure 8C2**).

## Other nuclei of interest

There were three particularly interesting findings regarding premotor nuclei labeled through our masseter and genioglossus rabies tracing experiments that deserve more detailed discussion.

The NTS (nucleus of the solitary tract) is thought to play a major role in taste processing (**Bradley and Grabauskas, 1998**). Previous tracing studies have suggested that the NTS contains neurons which project directly to the hypoglossal nucleus (**Aldes, 1980**; **Borke et al., 1983**; **Travers and Norgren, 1983**; **Dobbins and Feldman, 1995**; **Ugolini, 1995**; **Fay and Norgren, 1997**). Our results support these findings, revealing that some NTS neurons provide monosynaptic input onto tongue-protruding genioglossus motoneurons (**Figure 4B**). While sparse, this input could play a central role in the well-established taste reactivity test (**Grill and Norgren, 1978b**), which has been used extensively in assessing the interaction of taste processing and ingestive behaviors (**Flynn, 1995**). These stereotypic responses to controlled tastes have been found to be independent of cortical areas, and highly consistent across animals, suggesting that there may be a brainstem substrate for these behaviors (**Grill and Norgren, 1978a**). Our finding of a direct innervation of tongue motoneurons by NTS neurons may be one such neural substrate.

The labeling of neurons in the DCN (deep cerebellar nuclei) was somewhat surprising. The DCN had not previously been implicated in directly contacting masseter or genioglossus motoneurons of the brainstem in the neonate. Furthermore, where DCN axons project into the brainstem in non-murine species, the projection path is known to be primarily ipsilateral, with only sparse contralateral projections (**Cohen et al., 1958**; **Ruigrok and Voogd, 1990**). Previous retrograde or transsynaptic tracing studies in rodents did not comment on any deep cerebellar nuclei labeling, even in polysynaptic tracing experiments (**Travers and Norgren, 1983**; **Fay and Norgren, 1997**; **Kolta et al., 2000**). By contrast, in our transsynaptic studies we found that the fastigial nucleus of the DCN provides inputs onto contralateral masseter motoneurons in the early postnatal mouse (**Figure 2C**). Additionally, we observed that a small number of neurons in both the fastigial and dentate nuclei of the DCN provide inputs onto genioglossus motoneurons (**Figure 4D**, **Video 2**). Recently, the Allen Brain Institute published their adult mouse brain connectivity study (**Oh et al., 2014**). We consulted their connectivity atlas for tracing from the fastigial and dentate nuclei of the DCN. Interestingly, fastigial neurons give rise to a major contralateral projection pathway (as well as ipsilateral projections) and sparse fastigial axon terminals can be seen inside the contralateral MoV (**Allen Mouse Brain Connectivity Atlas, 2014a**; **Oh, et al., 2014**). Furthermore, tracing results from either the fastigial or the dentate nucleus revealed sparse axon terminals inside MoXII (**Allen Mouse Brain Connectivity Atlas, 2014b**; **Oh, et al., 2014**). Although these DCN innervations of contralateral MoV and MoXII are sparse in adult mice, it is possible that in newborn mice there are transiently more connections. Notably, in our P8→P15 masseter tracing experiment, we did not observe any labeling in the DCN, supporting such transient projections from DCN to masseter and genioglossus motoneurons during early development.

Finally, labeling of bilaterally projecting MesV primary afferent neurons, although very few, was another unexpected finding. In our P8→P15 tracing study, we only observed ipsilateral MesV neurons labeled. Thus, such bilaterally projecting MesV neurons could be another case that only transiently exists in neonatal animals.

In summary, our study revealed a set of premotor connection configurations well-suited to enable multi-muscle coordination and bilateral symmetry observed in feeding behaviors. Shared premotor neurons simultaneously providing inputs onto multiple groups of motoneurons innervating specific muscles may be a common circuit mechanism for motor coordination. It would be interesting to examine how these specific connections are established during development, and whether sensory inputs and CPG inputs converge on these coordinating premotor neurons to control both reflexes and

centrally-induced movements. Additionally, we provide high-resolution maps of premotor nuclei for these jaw and tongue motoneurons including sensory-related inputs and descending inputs. An important goal of future studies is to develop molecular tools that allow specific functional manipulation of individual groups of premotor neurons to establish their roles in controlling and coordinating a variety of orofacial behaviors.

## Materials and methods

All experimental protocols were approved by the Duke University Institutional Animal Care and Use Committee.

### Animals and tracing paradigm

We employed a previously described monosynaptic rabies-virus-based technique (*Wickersham et al., 2007a*; *Callaway, 2008*; *Arenkiel and Ehlers, 2009*) adapted to selectively trace neurons that directly synapse onto primary motoneurons (*Stepien et al., 2010*; *Takatoh et al., 2013*). Briefly, we used a knock-in mouse line containing CAG-loxP-STOP-loxP-rabies-G-IRES-TVA at the Rosa26 locus (RΦGT) (*Takatoh et al., 2013*) crossed with a mouse line that expresses Cre recombinase under the control of the choline acetyltransferase (ChAT) gene (Chat::Cre mouse, JAX Stock #006410). The resultant male and female pups were heterozygous for both alleles (Chat::Cre;RΦGT), and thus expressed the rabies glycoprotein (rabies-G) in all cholinergic neurons, including motoneurons (*Figure 1B*). Injection of a rabies virus expressing a fluorophore in place of the glycoprotein (green: ΔG-RV-EGFP; or red: ΔG-RV-mCherry) (*Wickersham et al., 2007b*) into the primary muscle (jaw-closing masseter or tongue-protruding genioglossus, *Figure 1A*) infected motoneurons targeting that muscle (*Figure 1C*, left). In the case of the masseter injections, animals which showed infection of the overlying skin as evident by fluorescence were excluded from analysis. Subsequent complementation of this virus by rabies-G in motoneurons enabled transsynaptic, retrograde travel of the virus into presynaptic partners (*Figure 1C*, right).

The monosynaptic rabies virus is deficient for the rabies-G in its genome, resulting in an inability for the virus to spread into presynaptic neurons of the source-infected cells unless it is complemented (*Etessami et al., 2000*). Additionally, rabies virus infects neurons potentially through binding to the neuronal cell adhesion molecule (NCAM) or other neuronal receptors irrespective of neurotransmitter phenotype, size, or morphology, enabling an unbiased assessment of presynaptic populations (*Thoulouze et al., 1998*; *Ugolini, 2010*).

As a caveat to our premotor tracing strategy, if any premotor neurons expressed ChAT, rabies virus could be complemented again, resulting in spurious two-step labeling. A previous study has reported small numbers of cholinergic neurons in the intermediate reticular region of the brainstem that project to MoV and MoXII (*Travers et al., 2005*). To investigate whether there were any ChAT+ masseter or genioglossus premoter neurons labeled in our experiments, we conducted anti-ChAT immunostaining on samples infected with rabies. We found that only approximately 1–3 cells in the reticular regions in the samples we examined were labeled (*Figure 1—figure supplement 1*). Additionally, such rare labeling was inconsistent in terms of rostral–caudal position across samples, suggesting that secondary jumping from such neurons would be very sparse with inconsistent labeling patterns. We therefore focused our quantification and analysis on the premotor regions consistently labeled across all samples.

### Viral injections

ΔG-RV was prepared and injected as previously described (*Takatoh et al., 2013*). Briefly, male and female mouse pups at postnatal day 1 (P1) were anesthetized by hypothermia, and were injected with 200–400 nL of ΔG-RV into either the masseter muscle or the genioglossus muscle of the tongue (*Figure 1A*). 1 week post-infection (*Figure 1C*, inset), mice were deeply anesthetized, transcardially perfused with 4% paraformaldehyde (PFA) in 1X phosphate-buffered saline (PBS), post-fixed overnight in 4% PFA, and cryoprotected in 30% sucrose solution in PBS. Brain samples were embedded in Optimal Cutting Temperature compound (OCT, Tissue-Tek) and frozen for at least 24 hours at −80°C. Optimal survival times for maximal labeling were assessed prior to collection of results. Periods longer than 1 week resulted in an increase in glial infiltration and cell debris in the motor nuclei suggesting massive cell death, while periods shorter than 5–6 days post-infection resulted in a significant decrease in transsynaptically labeled neurons.

## Histology

Tissue preparation and immunostaining were conducted as previously described (*Takatoh et al., 2013*). 80-μm thick sections were analyzed for tracing results, while ChAT immunostained samples were sectioned at 60 μm. Antibodies used were: goat anti-ChAT (1:1000, Millipore), rabbit anti-GFP (1:1000, Abcam), rabbit anti-RFP (1:1000, Rockland), Alexa488 anti-rabbit (1:1000, Jackson ImmunoResearch), Alexa594 anti-goat (1:1000, Jackson ImmunoResearch), Alexa488 anti-goat (1:1000, Jackson ImmunoResearch). In situ hybridization was performed as previously described with each probe being applied to every sixth 20-μm thick section (*Hasegawa et al., 2007*). GAD1, GAD2, GlyT2, vGluT2, and Tph2 probes were created as previously described (*Takatoh et al., 2013*). All probes were chosen based on whether they are known to be expressed in the brainstem regions of interest as seen in samples from the Allen Brain Atlas at www.brain-map.org. In addition to immunostaining and in situ hybridization, all sections were stained with DAPI (1:2000) to visualize the nuclei of all cells.

## Image Acquisition

Samples were imaged as previously described (*Takatoh et al., 2013*) using a Zeiss 710 inverted confocal microscope at 20X resolution. High resolution bouton images (*Figure 3G–H*; *Figure 5*) were obtained as 100X z-stacks. Full-field views of axon collaterals entering central facial, digastric, and contralateral trigeminal nuclei were obtained using a Zeiss inverted epifluorescent microscope at 10X resolution.

## Quantitative analysis

Infected premotor neurons in each of the brainstem areas were counted manually through serial sections. At least 5 animals were used for quantification. Final results were reported as percent of total premotor neurons labeled, thus normalizing each value to the overall infection level of that sample. Means were calculated from five samples each with primary infection of the masseter or genioglossus. All data are presented as mean percent of total neurons labeled ± standard error of the mean (SEM). Videos were generated using Adobe Photoshop CS6 to align all serial sections and export frames into an MP4 format. The videos were subsequently compressed using proprietary software.

## Acknowledgements

The authors would like to thank Drs Richard Mooney, Katsuyasu Sakurai, Rebecca Yang, Jeffrey Moore, Henry Yin, and Stephen Lisberger for their helpful advice during the process of this work and/or comments on this manuscript. This work was supported by grants from the NIH (NS077986 and DE019440) to FW, and FDE024003A to ES. The authors declare no competing financial interests.

## Additional information

### Funding

| Funder | Grant reference number | Author |
| --- | --- | --- |
| National Institutes of Health | NS077986 | Fan Wang |
| National Institutes of Health | DE019440 | Fan Wang |
| National Institutes of Health | FDE024003A | Edward Stanek IV |

The funders had no role in study design, data collection and interpretation, or the decision to submit the work for publication.

### Author contributions

ES, Conception and design, Acquisition of data, Analysis and interpretation of data, Drafting or revising the article, Contributed unpublished essential data or reagents; SC, Acquisition of data, Analysis and interpretation of data; JT, Analysis and interpretation of data, Drafting or revising the article, Contributed unpublished essential data or reagents; B-XH, Acquisition of data, Contributed unpublished essential data or reagents; FW, Conception and design, Analysis and interpretation of data, Drafting or revising the article

### Ethics

Animal experimentation: This study was performed in strict accordance with the recommendations in the Guide for the Care and Use of Laboratory Animals of the National Institutes of Health. All of the

animals were handled according to approved institutional animal care and use committee (IACUC) protocols (#A220-12-08) of Duke University. Duke University is fully accredited by the Association for Assessment and Accreditation of Laboratory Animal Care International (AAALAC).

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
