## [Decision Letter]

Thank you for sending your work entitled “Monosynaptic premotor circuit tracing reveals neural substrates for oro-motor coordination” for consideration at *eLife*. Your article has been favorably evaluated by a Senior editor and 3 reviewers, one of whom, Peggy Mason, is a member of our Board of Reviewing Editors.

The Reviewing editor and the other reviewers discussed their comments before we reached this decision, and the Reviewing editor has assembled the following comments to help you prepare a revised submission.

This is an important and interesting study that uses an elegant approach to investigate the brainstem premotor circuitry responsible for oro-motor coordination. Transgenic mice expressing rabies glycoprotein (G) in cholinergic neurons received injections of rabies virus expressing a fluorophore (in place of G) in their masseter muscles and/or genioglossus muscles. This resulted in retrograde labeling of premotor neurons projecting directly to the motoneurons of these muscles. Many of the findings reported here reproduce or confirm previously reported results, but innovate in 2 ways. Firstly, as pointed out by the authors, the methods used eliminate the possibility that some of the retrograde labelling results from having the tracer taken up by fibers passing by the injection site. Secondly, they allow direct comparison of the distributions of masseteric and genioglossus premotor circuitries. Several novel findings emerge from this study: premotor neurons have bilateral projections to masseteric motoneurons; premotor neurons innervate both the jaw and tongue muscles and are intermingled in the same regions of the reticular formation; a strong premotor innervation of genioglossus motoneurons from NTS. In summary, the findings provide evidence that the connectivity of premotor neurons might underlie coordination of orofacial muscles.

While the reviewers were unanimous that this manuscript tells an exciting, interesting and original story, the manuscript could be further strengthened by discussion and re-framing of several issues as follows: 1) At the age that the experiments were done, P1→P8, there is only suckling and swallowing, no chewing. The circuitry for ingestion is likely to change between the time studied (P1→P8) and the emergence of chewing. Indeed, the authors' previous paper in Neuron shows that vibrissal premotor circuitry changes extensively with emergence of exploratory whisking. The results should be interpreted in this light. And the authors should refer to the suckling CPG rather than the masticatory CPG. The labeling observed appears to be perfectly suited to suckling – opening mouth and lips to latch on, tasting milk, sucking, and swallowing.

2) The authors use “synchronization” and “coordination” loosely (Synchronization is maintenance of relative phase. Coordination involves modulation of muscle amplitude or timing (where modulation can be maintenance of amplitude or relative phase or changes in amplitude or relative phase) in order to achieve some goal. For example, during stage 1 food transport, jaw and tongue muscles are very differently coordinated than during mastication, but still well coordinated, and show variation in relative phase. In contrast, during mastication, jaw and tongue muscles are pretty well synchronized, but, again, their amplitude and relative phase of activity are constantly modulated to achieve the goal of chewing efficiently without biting the tongue. Thus, coordination and synchronization are not the same. These terms should be defined and used consistently and correctly.

3) Additional information about the methodology should be provided. Remember that there are no length constraints in *eLife*. Our hope is that all the information needed by a reader to understand what was done is contained in the Methods section. Therefore, please address the following methodological issues: a) How can the authors ensure that only monosynaptic projections are labeled? This is particularly worrisome in light of the cerebellar labeling. The adult cerebellum does not contain projections from DCN to motoneurons, even motoneurons in the brainstem. The cerebellar DCN labeling raises doubt that neurons are traveling retrogradely across a single synapse from the motoneuron.

b) Does the virus infect all cells equally independently of their size, metabolism or activity level?

c) How was the survival time determined? What are the consequences of having a too short or too long survival time?

d) Please motivate the use of VGluT2 (rather than VGluT1 or VGluT3) to mark glutamatergic neurons.

e) Is the virus injected through the skin when injected in the masseters? If yes, this should be stated clearly in the methods and the authors should consider the fact that some facial motoneurons (and their premotor connections) will be infected with the virus as well in this case.

4) The input from NTS to genioglossus but not masseter premotor neurons is striking. It is also very important as it is a potential substrate for the “taste reactivity” responses described by Grill and Norgren. These responses are innate, hardwired for particular tastes (modifiable but hard-wired to start with), independent of forebrain, and are used extensively to study liking vs wanting in rodents. The circuitry outlined here is very exciting in light of the importance of taste reactivity to diverse behavioral studies. The NTS premotoneuronal input to genioglossus is such an important finding and really should be included in the in the Discussion.

5) The cerebellar labeling is puzzling on a few accounts. In addition to #3a above, the masseter labeling is contralateral. We just don't understand this. While the data are what the data are, this is a puzzle that is worth noting. If the authors have any speculation on what is going on here, they should share it. In addition, genioglossus injections caused a very lateral labeling pattern (surely dentate) in the cerebellum. This is strange as one would expect labeling in the vermis and not in the lateral lobes. Again, the data are what the data are but these unexpected observations should be discussed.

---

## [Author Response]

1*) At the age that the experiments were done, P1→P8, there is only suckling and swallowing, no chewing. The circuitry for ingestion is likely to change between the time studied (P1→P8) and the emergence of chewing. Indeed, the authors' previous paper in Neuron shows that vibrissal premotor circuitry changes extensively with emergence of exploratory whisking. The results should be interpreted in this light. And the authors should refer to the suckling CPG rather than the masticatory CPG. The labeling observed appears to be perfectly suited to suckling – opening mouth and lips to latch on, tasting milk, sucking, and swallowing*.

We thank the reviewers for pointing out that the neonatal tracing (P1→P8) likely represents suckling premotor circuitry as chewing appears later in life. We have accordingly pointed this out in the Results section We also cited previous studies that found that over the time period when chewing behavior develops, glycine switches from being excitatory onto motoneurons to being inhibitory (28; 48), suggesting the possibility that the same circuitry used to produce rhythmic suckling early in life may be reconfigured to produce rhythmic chewing later in development (36; 74; 45), although this issue is still up for debate.

Furthermore, we conducted rabies-mediated transsynaptic tracing at a later developmental stage (P8→P15 tracing). Since chewing movements first emerge around post-natal day 12 (74), sampling at P15 should reveal chewing related premotor circuitry. These results are shown in the newly added supplemental figure (Figure 2—figure supplement 1).

For the P8→P15 tracing, we observed a much lower efficiency of viral infection of motoneurons (19±4 motoneurons labeled in P8 injected animals, as compared to 35±6 motoneurons labeled in P1 injected animals; mean ± SEM, n=3 samples per group), and thus the overall number of labeled premotor neurons was drastically reduced. Despite this, the labeled neurons were distributed in similar locations as those observed in the P1→P8 tracing (Figure 2—figure supplement 1). Interestingly, in contrast to other regions, labeling of premotor neurons in the LPGi was increased (compare Figure 2—figure supplement 1 to Figure 2). This is similar to that observed in our lab’s previous work by [62], except that in the whisker premotor circuitry, LPGi labeling transitioned from none to many across development, whereas for masseter premotor circuitry, LPGi labeling transitioned from few to many across development (see Figure 2—figure supplement 1). It will be interesting in the future to functionally test the role of LPGi neurons presynaptic to different cranial motor neurons. Due to the low infection and low spreading efficiency, we did not further pursue P8→P15 tracing for other experiments. This explanation has been added to the Results section.

*2) The authors use “synchronization” and “coordination” loosely (Synchronization is maintenance of relative phase. Coordination involves modulation of muscle amplitude or timing (where modulation can be maintenance of amplitude or relative phase or changes in amplitude or relative phase) in order to achieve some goal. For example, during stage 1 food transport, jaw and tongue muscles are very differently coordinated than during mastication, but still well coordinated, and show variation in relative phase. In contrast, during mastication, jaw and tongue muscles are pretty well synchronized, but, again, their amplitude and relative phase of activity are constantly modulated to achieve the goal of chewing efficiently without biting the tongue. Thus, coordination and synchronization are not the same. These terms should be defined and used consistently and correctly*.

We thank the reviewers for the clarification, and have changed the use of these words accordingly throughout the manuscript.

*3) Additional information about the methodology should be provided. Remember that there are no length constraints in eLife. Our hope is that all the information needed by a reader to understand what was done is contained in the Methods section. Therefore, please address the following methodological issues: a) How can the authors ensure that only monosynaptic projections are labeled? This is particularly worrisome in light of the cerebellar labeling. The adult cerebellum does not contain projections from DCN to motoneurons, even motoneurons in the brainstem. The cerebellar DCN labeling raises doubt that neurons are traveling retrogradely across a single synapse from the motoneuron*.

We apologize for not describing the methods more thoroughly, and all questions are great questions. We have included detailed description (methodological description, references, as well as some results) to address all the questions (a-e) raised by the reviewers.

Could there be secondary jumping of the rabies virus in our approach? This is a very good concern. In our experiments, complementation of the virus is limited to occurring solely in cholinergic neurons, which express the rabies-G induced by the ChAT-Cre transgene. While this primarily limits complementation to within motoneurons, there are also central cholinergic neurons located in the reticular regions of the brainstem, and a previous retrograde tracing study found that some of these neurons project to the MoV and MoXII (70). When we performed anti-ChAT staining on the viral-labeled brain samples, we observed occasional and rare labeling of cholinergic premotor neurons located in the reticular formation (about 1∼3 per brain). This is now described in detail in the Methods section as a caveat, and the representative results of the anti-ChAT staining are shown in Figure 1—figure supplement 1.

These rare labeled central cholinergic premotor neurons in theory could enable the rabies virus to be complemented again and travel one more step to pre-premotor neurons. However, the location of these rarely-labeled neurons were highly variable across the brainstem between samples, suggesting that even if jumping occurred, the labeled pre-premotor neurons may not exhibit a consistent labeling pattern (in terms of both number and location of presynaptic contacts). In addition, the brains were sampled 7 days after initial viral infection. Because it takes time for the virus to replicate and be complimented by the rabies glycoprotein, the chance of secondary “jumping” is further diminished. Nevertheless, we indeed observed some sparse inconsistent labeling in various regions of brainstem (as mentioned in the results). Our quantitative analyses (Tables 1 and 2) were performed only on consistently labeled brainstem regions.

Problems with cerebellar DCN labeling: the reviewers are absolutely right with the concerns of cerebellar DCN labeling. First of all, we should have been more precise about where we observed the labeling in the DCN. The masseter premotor study consistently labeled neurons in the contralateral *fastigial* nucleus of the DCN, whereas the genioglossus premotor study sparsely labeled neurons in the ipsilateral *fastigial* and *dentate* nuclei of the DCN. We looked into this issue further. We are fortunate that the Allen Brain Institute just published their “Mouse Brain Connectivity Atlas” which included anterograde tracing studies from the fastigial and dentate nuclei of the DCN. Importantly, fastigial neurons send a major projection *contralaterally* in addition to an ipsilateral projection, and a few axons can be seen projecting into the contralateral MoV even in the adult mouse (see images below, Figure 9, obtained from the Allen Institute at connectivity.brain-map.org). Furthermore, a few axons from both the fastigial and dentate nuclei can be observed inside the MoXII (see images below from the Allen Institute at connectivity.brain-map.org). These sparse innervations in the adult raise the possibility that in neonatal mice there is a transient more elaborate projection from DCN to cranial motor nuclei that is later largely pruned away, but was detected by our P1→P8 transsyanptic tracing. Additionally, we noted that our late-stage P8→P15 tracing did not label any neurons in DCN, consistent with a transient synaptic contact between DCN neurons and masseter or genioglossus motoneurons at early neonatal stage. This explanation has been added to the Discussion.Author response image 1.

*b) Does the virus infect all cells*
*equally independently of their size, metabolism or activity level?*

We added references about primary infection and spreading of the rabies virus in the Methods. Briefly, both the infection and spreading of rabies virus are exclusively dependent on the rabies-G protein (17). Rabies-G binds to synapse-localized molecules, such as the neuronal cell adhesion molecule (NCAM) (67), after which the virus is internalized through endocytosis. Studies of rabies viral infection have shown a lack of preference in transmission for neurotransmitter phenotype or cell body distance from initially infected neurons, and rabies has been shown to infect neurons of various cell soma sizes and activity levels, as reviewed by Gabriella Ugolini (72). Description of the rabies virus has now been included in the Methods.

*c) How was the survival time determined? What are the consequences of having a too short or too long survival*
*time?*

The survival/sampling time was primarily determined by the toxicity of the rabies virus and the extent of transsynaptic labeling. When sampling at shorter time points post-infection (4∼5 days), we observed significantly fewer labeled premotor neurons. When sampling at time points longer than one week post-infection (8∼9 days), we see an increase in glial cell infiltration in the motor nucleus, loss of labeled motor neurons and presence of cell debris suggesting that rabies has caused death of infected neurons. This explanation has now been included in the Methods.

*d) Please motivate the use of VGluT2 (rather than VGluT1 or VGluT3) to mark glutamatergic neurons*.

The primary reason for using vGluT2 instead of vGluT1 or vGluT3 to examine transmitter phenotypes of labeled premotor neurons was based on the expression patterns of these three genes as shown in the Allen Brain Atlas. While vGluT1 is an important marker for MesV neurons, we were primarily interested in assessing neurotransmitter phenotypes in the reticular formation, where vGluT2 is extensively expressed. By contrast, vGluT1 is only expressed in small number of neurons and largely absent in reticular formation prevalent, and vGluT3 is expressed in very few neurons in the brainstem. We have now included this rationale for the probes used in the Methods.

*e) Is the virus injected through the skin when injected in the masseters? If yes, this should be stated clearly in the methods and the authors should consider the fact that some facial motoneurons (and their premotor connections) will be infected with the virus as well in this case*.

We apologize for not providing enough detail with regards to peripheral injection. Yes, the virus is injected through the skin into the masseter muscle. Hence there was the possibility of the virus leaking out and infecting facial motor axons passing through underneath the skin. When harvesting the mouse after tracing, we examined the tissue at the injection site under a fluorescence microscope, and excluded samples that showed infection of the dermal and epidermal tissues. Additionally, we did not observe any ectopic motoneurons labeled by rabies in the facial motor nucleus or any other cranial motor nuclei when we examined our serial sections. We have now included this explanation in the Methods.

*4) The input from NTS to genioglossus but not masseter premotor neurons is striking. It is also very important as it is a potential substrate for the “taste reactivity” responses described by Grill and Norgren. These responses are innate, hardwired for particular tastes (modifiable but hard-wired to start with), independent of forebrain, and are used extensively to study liking vs wanting in rodents. The circuitry outlined here is very exciting in light of the importance of taste reactivity to diverse behavioral studies. The NTS premotoneuronal input to genioglossus is such an important finding and really should be included in the in the Discussion*.

We appreciate the reviewers’ insight on the importance of this finding. We have added summary sentences according to reviewers’ suggestion. We also included a magnified image of the NTS labeling in Figure 4, and have further discussed it in the Discussion citing these references mentioned by the reviewers.

*5) The cerebellar labeling is puzzling on a few accounts. In addition to #3a above, the masseter labeling is contralateral. We just don't understand this. While the data are what the data are, this is a puzzle that is worth noting. If the authors have any speculation on what is going on here, they should share it. In addition, genioglossus injections caused a very lateral labeling pattern (surely dentate) in the cerebellum. This is strange as one would expect labeling in the vermis and not in the lateral lobes. Again, the data are what the data are but these unexpected observations should be discussed*.

The reviewers are absolutely right about the surprising and somewhat puzzling labeling of premotor neurons in cerebellar nuclei. Previous tracing experiments have found some contralateral projections from the deep cerebellar nuclei in the cat (15) and rat (55). And as mentioned above (see response to #3a), recent anterograde tracing experiments conducted by the Allen Institute in the adult mouse from the fastigial nucleus show a major contralateral projection pathway with a few axons innervating the contralateral MoV (see attached images from connectivity.brain-map.org). Thus, it appears that the mouse has a more prominent contralateral projection from fastigial nucleus. Additionally, sparse processes are also visible extending into the MoXII from both the fastigial and dentate tracing by the Allen Institute. It is possible that the deep cerebellar nuclei are connected to motoneurons only temporarily during development (see response to 3a). We have included extended discussion on this labeling pattern in the Discussion, as per the reviewers’ suggestion.